# A Systematic Literature Review on PHM Strategies for (Hydraulic) Primary Flight Control Actuation Systems

**Leonardo Baldo \*** , **Andrea De Martin** , **Giovanni Jacazio** and **Massimo Sorli**

Department of Mechanical and Aerospace Engineering, Politecnico di Torino, Corso Duca degli Abruzzi 24, 10129 Torino, Italy; andrea.demartin@polito.it (A.D.M.); giovanni.jacazio@formerfaculty.polito.it (G.J.); massimo.sorli@polito.it (M.S.)
* Correspondence: leonardo.baldo@polito.it

**Abstract**

Prognostic and Health Management (PHM) strategies are gaining increasingly more traction in almost every field of engineering, offering stakeholders advanced capabilities in system monitoring, anomaly detection, and predictive maintenance. Primary flight control actuators are safety-critical elements within aircraft flight control systems (FCSs), and currently, they are mainly based on Electro-Hydraulic Actuators (EHAs) or Electro-Hydrostatic Actuators (EHSAs). Despite the widespread diffusion of PHM methodologies, the application of these technologies for EHAs is still somewhat limited, and the available information is often restricted to the industrial sector. To fill this gap, this paper provides an in-depth analysis of state-of-the-art EHA PHM strategies for aerospace applications, as well as their limitations and further developments through a Systematic Literature Review (SLR). An objective and clear methodology, combined with the use of attractive and informative graphics, guides the reader towards a thorough investigation of the state of the art, as well as the challenges in the field that limit a wider implementation. It is deemed that the information presented in this review will be useful for new researchers and industry engineers as it provides indications for conducting research in this specific and still not very investigated sector.

**Keywords:** electro-hydraulic actuators; flight controls; prognostics and health management; systematic literature review; PICOC; PRISMA

## 1. Introduction

The onset of Industry 4.0 and the widespread use of Industrial Internet of Things (IIoT) have deeply affected the way component and subsystem operating data can be handled: a large amount of data can now be processed in an optimized way, thus providing engineers with the capability of monitoring component health status and operating behavior. In fact, data can then be turned into information, information into knowledge, and knowledge into decisions through a Data-to-Decision process [1]. This on-the-edge data management, together with aircraft electrification linked to the More Electric Aircraft (MEA) paradigm [2], has been a key enabler for Electro-Mechanical Actuators (EMAs) Prognostic and Health Management (PHM) strategies, which represent a very popular research topic [3,4].

PHM methodologies can be seen as a combination of several interconnected functional layers: the diagnostic layer handles failure detection, isolation, and quantification; the prognostic layer focuses on performance assessment and prediction of the system Remaining Useful Life (RUL); and the health management layer, which leverages this information to

optimize asset operations [5]. Since the implementation of PHM strategies in the industrial and aerospace sectors, numerous systems have been the focus of research aimed at developing effective prognostic strategies. Kordestani et al. [6] have carried out an extensive and detailed overview of some areas where PHM is used in aeronautical applications, as well as the related challenges and opportunities. In particular, the study focused on turbofan engines, Multi-Functional Spoilers (MFSs), and electro-hydraulic servo valves. Some other prominent areas where PHM solutions are already developed in the aerospace sector are landing gear brakes [7], lithium-ion batteries [8,9], and other subsystems where data are easily accessible (e.g., bearings or brake disk degradation) and/or a large amount of data are already monitored (e.g., Full Authority Digital Engine Control (FADEC) systems) and/or features are relatively easy to monitor (lithium-ion batteries). Furthermore, research in these fields is supported by a comprehensive set of available databases that have, over the years, become the standard reference starting point (e.g., C-MAPSS [10], NASA Battery Dataset [11], NASA Bearings Dataset [12]). This is the case of turbofan engines, where the research line is propelled forward thanks to the availability of the C-MAPSS prognostic dataset [13–15], which addresses the engine's overall state of health. Some studies even try to create a single prognostic system involving more subsystems [16]. PHM-positive outcomes influence a wide range of operating domains, starting from a mere reliability and safety standpoint to customized Integrated Logistic Support (ILS) or Performance-Based Logistics (PBL) chains [17]. Opportunistic [18,19], predictive [20,21], prescriptive [22] and Condition-Based Maintenance (CBM) [23,24] concepts can lay their roots upon PHM strategies leading to operating cost cutbacks, availability improvements, and to a seamless asset management, bringing positive impacts to both the OEM and the operators [25]. Very little material can be found in the literature for PHM for primary flight controls, and a few research groups from a limited number of universities and research clusters are focusing on this issue. On the other hand, with a striking comparison, Electro-Hydraulic Actuators (EHAs) can be found on every commercial aircraft and constitute the backbone of flight actuation systems, thanks to their time-proven, robust, and reliable technologies and, as such, require interest [26]. On top of that, cutting-edge approaches related to digital twins [18,27,28], Behavioral Digital Aircraft, and even Integrated Vehicle Health Management (IVHM) [29] systems based on the current generation of aircraft need stable and reliable PHM systems for EHAs. That is why, to fill this gap, the authors strongly believe that a review of the actual state of the art is extremely beneficial for the PHM community and the overall aerospace sector, providing a foundation and a reference for future PHM-related activities. While it is true that failures affecting the various sub-systems that interact with the actuators may reflect an actuator-level performance degradation, this review specifically focuses on actuator-level failures. Given the topic relevance, an in-depth Systematic Literature Review (SLR) on PHM for primary flight control actuators with specific queries has been carried out, looking at the technology trends in the latest years and highlighting the most influential contributors and the leading sources. After a brief introduction on EHAs and flight control systems in general, the most updated methods have been compared following a systematic, objective, and reproducible rationale. The remainder of this paper is organized as follows: Section 1.1 provides essential background information, introducing flight control systems (FCSs), the key differences between primary and secondary flight controls, and Electro-Hydrostatic Actuators (EHAs), along with an overview of high-level PHM implementation approaches. Section 2 presents the SLR protocol and methodology, with a focus on the search strategy, inclusion/exclusion criteria, research questions, and the overall framework employed in this study. Section 3 addresses the highlighted research questions, examining the current state of the art, leading research contributors, methodological approaches, commonly investigated components and fault

modes, used signals, and implementation challenges. Finally, Section 4 synthesizes the findings and provides insights on future research and industrial applications.

### 1.1. Flight Control Computers, Electro-Hydraulic Actuators, and PHM
#### 1.1.1. FCSs and Electro-Hydraulic Actuators

Flight control refers to the movable aerodynamic surfaces and systems employed to manage the aircraft movement during flight. FCSs encompass the set of components necessary to convey flight control commands from the cockpit or other sources (e.g., the autopilot or trim systems through the Flight Control Computer (FCC)) to the appropriate actuators, producing forces and torques that determine the aircraft's performance and controllability in terms of attitude, airspeed, flight path, and position [30]. To better understand the roles and safety relevance of these components, flight controls are typically divided into primary and secondary systems. Primary flight controls are critical for the safe operation of an aircraft and, in conventional configurations, include the ailerons, elevator (or stabilator), and rudder. These controls are flight-critical/safety-critical systems, and they are essential to control aircraft position, speed, and attitude. In contrast, secondary flight controls are employed to enhance performance and reduce the pilot workload. They often include devices such as flaps, slats, and trim systems. Finally, spoilers are flight control surfaces located on the aircraft wing top, serving two primary functions: enhancing roll control and acting as speed brakes. As a result, they can be classified as primary or secondary flight controls depending on the actual intended and designed use. Although loss of secondary controls can complicate aircraft handling and reduce performance or safety margins in certain flight phases, it does not inherently compromise basic controllability. Therefore, the focus of safety-critical design and monitoring is predominantly on the primary flight controls. In order to move the aerodynamic surfaces and provide the aforementioned forces and torques, the FCS needs actuators. The actuators employed in primary flight controls differ substantially from those used in secondary flight controls due to the peculiarities and design requirements of each application. For instance, primary flight controls necessitate smooth, continuous deflections of the control surfaces to enable precise and responsive maneuvering of the aircraft throughout all flight phases. In contrast, secondary controls such as flaps operate with discrete deflection settings. Flap positions are typically selected by the pilot via a control knob with a limited number of preset positions, resulting in only a few available final configurations. This distinction is reflected in the design of the actuators used for primary and secondary flight controls. For example, rotary actuators are frequently utilized in secondary flight control systems, whereas linear actuators are typically employed in primary flight controls. Moreover, the transition to electrically powered actuators is particularly significant for secondary flight controls. However, for primary flight controls, linear EHAs remain the preferred choice due to their proven reliability, ruggedness, and high power density. For safety-critical applications, EHAs are usually configured in an active–standby tandem configuration, where the standby cylinder helps to damp external oscillations and vibrations. Due to the substantially different configurations and design requirements of actuators for primary and secondary flight controls, a unified treatment of both types would lead to a loss of clarity and hinder a clear understanding of the employed PHM strategies. Consequently, as outlined in the review protocol presented in Section 2, this review focuses exclusively on primary flight control actuators and, accordingly, considers only linear EHAs.

It is quite difficult to describe a generic EHA architecture given the wide range of EHA variants. However, as reported by J. C. Maré [26], an overview of an EHA shows some common traits: the servo valve, which is the link between the hydraulic system and the electrical one, the main (linear) actuator, the control electronics, sensors, and transducers.

The servo valve is usually selected between three main types: a proportional direct-drive servo valve, a jet-pipe, or a flapper-nozzle one. Each one has its pros and cons [31,32]. Up to these days, the servo valve command can be transferred from the pilot's stick in mainly two ways: a mechanical link or by Fly-By-Wire (FBW) technology. The former solution requires a complicated set of mechanical pulleys and rods that link the output of the control stick to the servo-valve input interface [33]. This is a time-proven and effective solution employed when the control surfaces are not excessively large and when the distance between the pilot and the aerodynamic surfaces is not too extended. Moreover, this strategy is extremely cost-effective. On the other hand, in some platforms, as weight and airspeed started to become unbearable for a purely mechanical system, a higher degree of responsiveness, power, and digitalization was sought. This is where Fly-By-Wire (FBW) solutions came into place [34]. According to a low-power electrical signal, the servo valve drives the hydraulic flow (and power) in one of the two chambers of the hydraulic cylinder, hence providing the control desired by the user. The hydraulic system is connected to the servo valve through an intricate set of secondary valves (bypass, isolation, shut-off, etc.) and components (accumulators, filters, and sensors) to guarantee continuous hydraulic power and safety features. The low-power electrical signal is generated in the cockpit through transducers and sensors, which convert the pilot's movements on the control column (or stick) and send the signal to the FCC, which elaborates it. Finally, through FBW technology, the signal is sent to the EHAs' input connectors. These signals are now ready to be handled by the control electronics and the servo valve. The control logic is usually composed of a transducer, often a Linear or Rotary Variable Density Transformer (LVDT/RVDT), which feeds back the rod position and closes the position control loop typical of primary flight control surfaces. Other internal control loops may involve speed, spool position, or current. The control logic involves a digital controller (often a PI or PID controller), which compares the reference position and the actual position sensed by the LVDT. Additionally, in some configurations, there are some more sensors placed in the EHA itself, providing useful information about pressure, the presence of debris and metal particles, the position of the spool, etc. All these data, when logged in a system like a Health Usage and Monitoring System (HUMS), are pivotal for the development of PHM solutions for a very safety-critical system such as the FCS. Moreover, in a typical System-of-Systems (SoS) perspective, EHAs interact with a wide range of components and systems, such as pipelines, mechanical transmissions, and electronic wiring. In this context, Zhang et al. [35] and Ye et al. [36] developed a pipeline contamination and leakage prediction model, respectively. Kosova and Unver [37] presented a digital twin framework for the failure detection of hydraulic systems. Shen and Zhao [38] and Liu et al. [39] presented a fault analysis strategy for aircraft hydraulic systems. Yang et al. presented a review on diagnostic strategies for hydraulic pumps [40]. Zong et al. [41] presented a real-time monitoring system for the actuator mechanism of an aileron, focusing on the comparison between different dynamic responses. While failures on these subsystems may influence the EHA's overall behavior, the paper analyses only actuator-level failures.

### 1.1.2. An Overview of How PHM Can Be Performed

PHM approaches are often labeled as Data-Driven, Knowledge-Based, Physics-Based, Model-Based, and Hybrid methodologies [25,42–46].

- Data-Driven approaches are big data-focused techniques in which a large volume of historical data on the state of the asset is processed thanks to data analysis algorithms with different levels of explainability, Artificial Intelligence (AI) integration, and complexity [47]. As a result, useful information can be extracted from historical data to learn degradation trends and foresee the future health status without the need for

precise knowledge of the system. Data-Driven methods, as the name implies, require a strong and solid database to rely on, which can be extracted from past observations or from models and simulations. Usually, Data-Driven methodologies are quick to implement and less expensive to build and deploy, but the reliance on a vast dataset covering a wide range of operational conditions limits the actual effectiveness of these strategies to the few cases when an extensive dataset is available. Other issues are often linked to the scarce generalization ability of the developed algorithms, which hardly extend to conditions not covered in the dataset. Usually, aerospace systems are designed with very high safety requirements stated by the relevant certification specification and, as a result, the failures are very limited. This behavior, called the "few-shot phenomenon", leads to a very unbalanced dataset, with the faulty class being often underrepresented in the dataset. As a result, insufficient or imbalanced data may indeed jeopardize the strategy's performance. Therefore, data quality and quantity must be assessed prior to the methodology definition.

- Knowledge-Based and Model-Based approaches share a lot of similarities as they rely on precise and detailed information on the system being analyzed. These strategies are usually more accurate, explainable, and precise, but, on the other hand, they suffer from high cost, time consumption, intensive computational costs, and the need for accurate and in-depth system knowledge.

  - Knowledge-Based PHM employs a priori knowledge about the system and utilizes expert knowledge and domain expertise to forecast the future performance and health of systems or machinery by integrating established knowledge, guidelines, and insights into the system's physical properties, as well as historical performance data.

  - Physics-Based approaches, often referred to as Model-Based or Physics-of-Failure (PoF) approaches, leverage a comprehensive understanding of the underlying physical principles and dynamics of a system to forecast its future performance and health status. They are based on the use of an analytical model, able to describe the monitored system in a mathematical way. Models can be designed with different architectures depending on the required level of detail and hierarchy of the system being investigated. The models are established on physical equations and mathematical frameworks that describe the system's operation, taking into account variables such as material properties, structural dynamics, forces, vibrations, and other relevant physical phenomena. In particular, PoF refers to the modeling of the degradation process under examination.

- Finally, the Hybrid approach combines both Model-Based and Data-Driven approaches, keeping their advantages [48–51]. Some new initiatives in recent years involve physics-informed approaches in various neural network architectures, with the aim of integrating the strong Data-Driven behavior with the generalization and robustness quality of analytical processes [52,53]. Physics-informed machine learning integrates (noisy) data and mathematical models, combining them through neural networks or other Deep Learning (DL) strategies. The prior knowledge of general physical laws can be embedded in various ways: for instance, in the loss function of the DL algorithms, inside neurons, or as regularization agents. In this way, the physical knowledge of the system is intertwined with the powerful DL architecture in a Hybrid fashion [27,53–59].

Numerous papers, reports, and best practices outline potential workflows and steps for designing a PHM framework for complex systems. However, these guidelines must be significantly modified and tailored to fit the specific case study.

## 2. Systematic Literature Review Protocol and Methodology

Literature reviews are a powerful tool used to identify the actual state of the art and discover connections and trends, as well as uncover gaps to define a topic so that research questions may be informed and further research can be carried out starting from a solid base.

To mitigate the classical weaknesses of literature reviews (i.e., the lack of an explicit methodology and the subjectivity), the authors decided to opt for an SLR. In fact, the SLR is chosen to target the main characteristics of an unbiased literature review: transparency, transferability, and replicability of the work. The objective of this study is defined in the form of research questions, which have been formulated leveraging the PICOC criteria [60,61]. These criteria are often used in literature reviews in the medical field, but they are starting to appear in engineering and computer science reviews as well, as they offer a great starting point to concretize and write down feasible research questions. PICOC is a method utilized to outline the five key components of a research topic.

The acronym "PICOC" represents the following:

- Population: What is being studied?
- Intervention: What action or approach is being implemented?
- Comparison: What is being used as a comparison?
- Outcome: What objectives or improvements are being sought?
- Context: In what organization or circumstances is this occurring?

In particular, in this case:

- Population: EHA-powered primary flight controls actuators.
- Intervention: PHM strategies.
- Comparison: Existing prognostic techniques used to identify and predict faults in EHAs.
- Outcome: Availability increase and cost-effectiveness.
- Context: Commercial aviation sector.

As a result, a list of research questions has been formulated and answered in the following paragraphs:

1. RQ1: What is the state-of-the-art of PHM in EHAs for primary flight controls?
2. RQ2: What are the most prominent authors, affiliations, and geographic areas with the highest number of records?
3. RQ3: Which are the most used approaches (Data-Driven, Model-Based, Hybrid) for diagnosis and prognosis?
4. RQ4: Which are the most investigated components and fault modes?
5. RQ5: Which are the most commonly used signals?
6. RQ6: Which methods and techniques are the most used ones?
7. RQ7: What are the current challenges that prevent PHM solutions for primary flight controls from increasing the product availability and cost-effectiveness?

To mitigate the classical weaknesses of literature reviews [60], it is pivotal to clearly state inclusion and exclusion criteria (ICs and ECs), as reported in the following list.

- IC1: The study must be related to PHM for EHAs in flight control actuators (both for fixed wing and rotary wing).
- IC2: The articles must develop at least a prognosis methodology (e.g., diagnostic-only papers are not considered).
- IC3: The study must include full text (e.g., abstract-only papers are excluded).
- IC4: Articles with prognosis and diagnosis are included.

- IC5: No period limitations have been applied, as the PHM field is quite recent, and no limitations on the type, accessibility, or impact of the source have been implemented either.
- EC1: The analysis of PHM methodologies for EMAs and Electro-HydroStatic Actuators (EHSAs) [62] has been excluded.
- EC2: Rotary actuators are excluded since they are not employed in primary flight control actuation systems where linear actuators are required.
- EC3: The analysis of the possible integration of these strategies in more complex frameworks (e.g., maintenance and scheduling optimization) is also not taken into consideration. Even if they are very significant and key drivers for the development of PHM systems themselves, including these additional topics would make this review much less readable and would require a separate study, such as the one carried out by M. J. Scott et al. in [63].
- EC4: Articles with only the diagnostic layer are excluded.
- EC5: Articles not in English or not publicly available have been excluded.

Comprehensive research was conducted on the Elsevier (E) and IEEE Xplore (I) databases to identify valuable papers, books, reports, and documents that contribute to the field of PHM with a specified set of search prompts. Other databases, such as Springer, were excluded, as advanced search options do not provide the desired level of prompt customization. This list is based on a set of keywords, employed both together as well as isolated: "PHM", "Electro-Hydraulic Actuator", "Electro-Hydraulic Servo-Actuator", "Prognostics", "Performance Degradation Prediction", "Hydraulic Servo System", "Fault Detection, Diagnosis and Performance Assessment", "Aircraft". In fact, a common problem often highlighted in this field is the use of different terms for the same approach. For instance, Prognostic and Health Monitoring can be used interchangeably with Prognostic and Health Management, or Remaining Useful Life Prediction, or Fault Detection, Diagnosis and Performance Assessment, Fault Diagnosis and Prognosis (FDP), or even Diagnosis, Prognosis, and Health Management (DPHM), etc. The comprehensive list of search prompts is reported in Figure 1, along with the number of Total Records (TRs) for each prompt retrieved in the two databases. A total of 433 records (including duplicates) have been found (294 from Elsevier and 139 from IEEE Xplore). To ensure rigor and consistency, the study adhered to the Preferred Reporting Items for Systematic Reviews and Meta-Analyses (PRISMA) guidelines, which informed the selection of published articles included in this analysis. The PRISMA diagram is reported in Figure 2, highlighting the different phases (Identification, Screening, Eligibility, and Inclusion). As stated before, after a research with the search prompts reported in Figure 1, a total of 433 records were identified from the two databases. After removing duplicates, 224 records were retrieved. One hundred seventy-six results were excluded after a screening of the title and abstract following the inclusion and exclusion criteria. The remaining 48 articles' full texts were then assessed for eligibility, and 20 were excluded.

As a result, a total of 28 articles were included in the review; among them, 15 are journal articles, while 13 are conference papers, as shown in Figure 3a. The list of the records that passed the PRISMA selection process is reported in Table 1, along with the authors, the year of publication, and the reference number according to the reference section at the end of the article.

The field was found to be emergent, with 82% (23 out of 28) of all articles published in the last ten years. The historical trend is reported in Figure 3b. The first record is from 2004, and the last one was published in 2023, as this work was carried out in 2024. A tendency towards higher numbers can be seen; however, as already stated and then better explained in Section 3.1, the available publications on these topics are quite limited.

As expected, the conferences and journal main themes are oriented towards mechanical, aerospace, and fluid dynamics topics. In Figure 4, a bar graph of the conference names is reported: the Annual Conference of the PHM Society is the main source (six papers), while the other two papers are related to the IEEE Aerospace Conference. The complete and detailed list of the source type (Conference Paper/Journal), as well as the name of the conference or journal, is reported in Table A5 in Appendix A. Following the selection process, an "a posteriori" lexicon analysis was performed on the titles of the articles to identify the most influential terms used in the study titles. Consequently, after excluding grammatical conjunctions (e.g., "and" "for" "in" etc.), the words in the titles were counted and visualized in a bar plot, as illustrated in Figure 5. The three most frequently occurring words are "hydraulic" with 16 occurrences; "electro" with 12 occurrences; and "servo" with 11 occurrences. Additionally, terms such as "prognosis" "health," and "prognostics" appear among the top 10 most commonly used words. This post hoc analysis serves as a valuable tool for understanding the key terms, which can facilitate future searches for related articles in subsequent updates of this literature review or assist readers seeking a more in-depth exploration of a particular field.

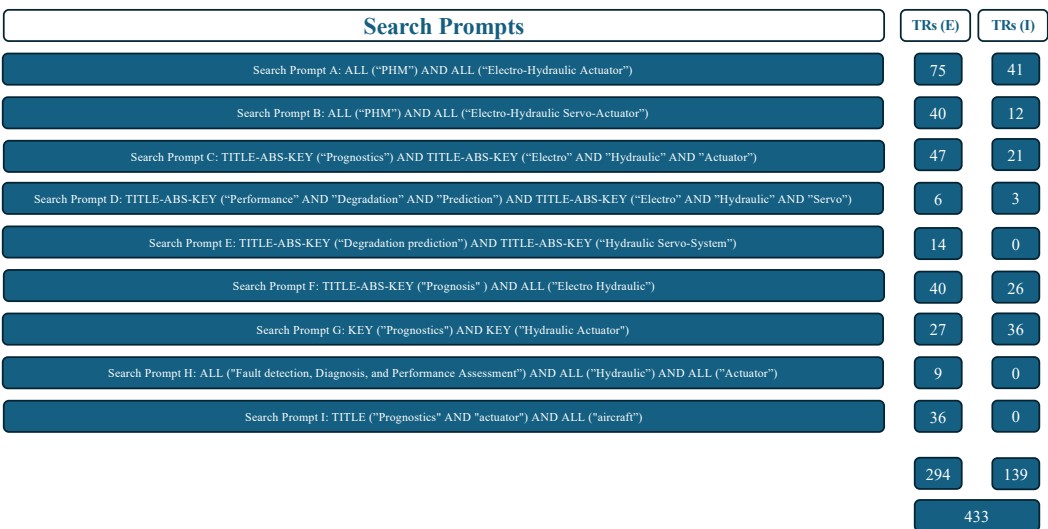

**Figure 1.** Search prompts and Total Records (TRs) retrieved for each prompt from the two selected databases.

**Table 1.** List of the 28 studies selected for this SLR.

| Authors | Year | Ref. No. |
|---|---|---|
| Byington et al. | 2004 | [64] |
| Byington et al. | 2004 | [65] |
| De Oliveira Bizzarria and Yoneyama | 2009 | [66] |
| Borello et al. | 2009 | [67] |
| Bartram abd Mahadevan | 2013 | [68] |
| Zhang et al. | 2014 | [69] |
| Liu et al. | 2015 | [70] |
| Bartram and Mahadevan | 2015 | [71] |
| Mornacchi et al. | 2015 | [72] |
| Zhenya et al. | 2015 | [73] |
| Soudbakhsh and Annaswamy | 2017 | [74] |
| Guo and Gan | 2017 | [75] |
| Macaluso and Jacazio | 2017 | [76] |
| Lu et al. | 2018 | [77] |
| Autin et al. | 2018 | [78] |

**Table 1.** *Cont.*

| Authors | Year | Ref. No. |
|---|---|---|
| Shahkar et al. | 2019 | [79] |
| Guo and Sui | 2019 | [80] |
| Guo and Sui | 2020 | [81] |
| Kordestani et al. | 2020 | [82] |
| Guo et al. | 2020 | [83] |
| Nesci et al. | 2020 | [84] |
| De Martin et al. | 2020 | [85] |
| Autin et al. | 2021 | [86] |
| Bertolino et al. | 2021 | [87] |
| De Martin et al. | 2022 | [88] |
| Shahkar and Khorasani | 2022 | [89] |
| Cui et al. | 2023 | [90] |
| Mi and Huang | 2023 | [91] |

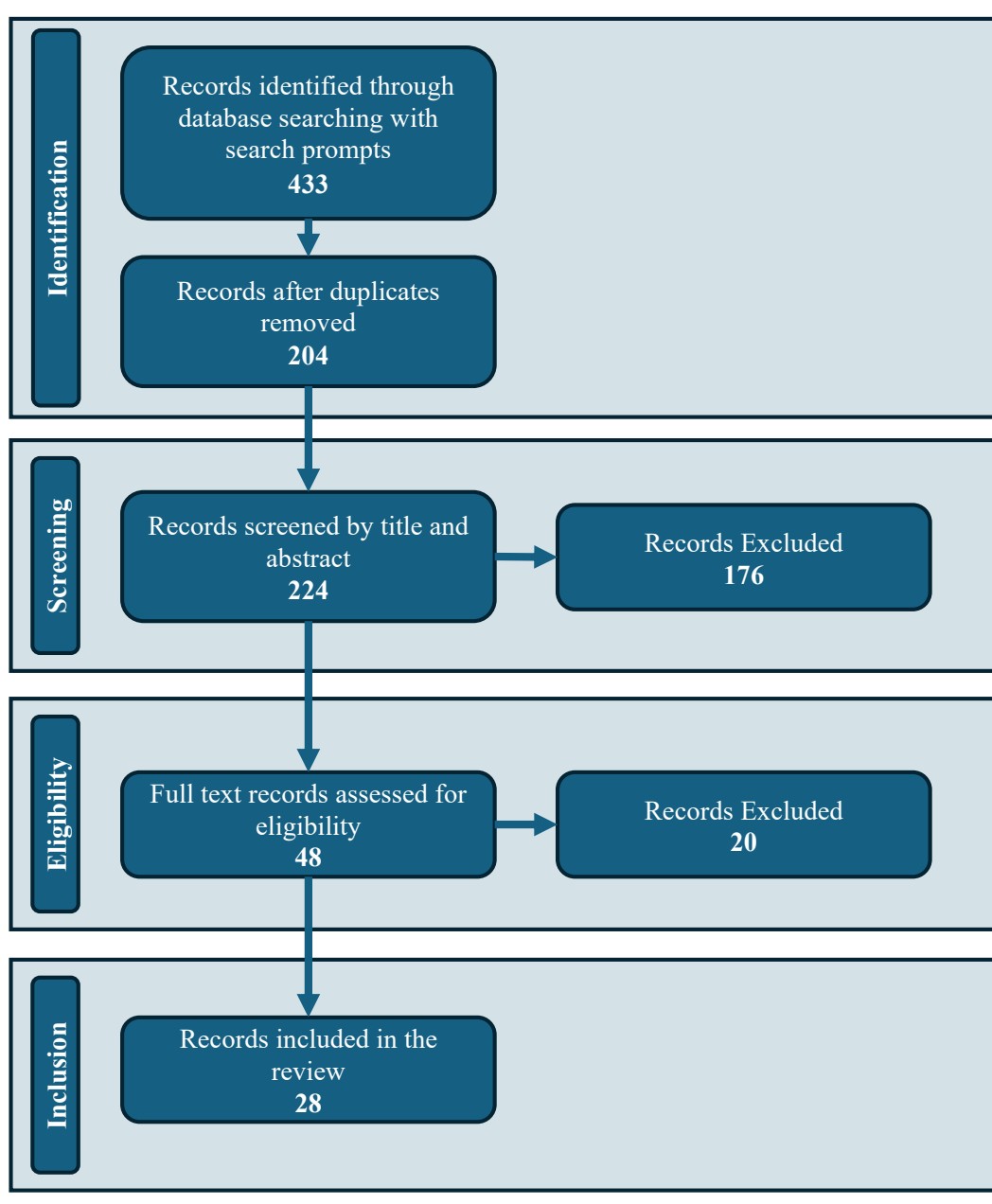

**Figure 2.** PRISMA diagram with record numbers and phases.

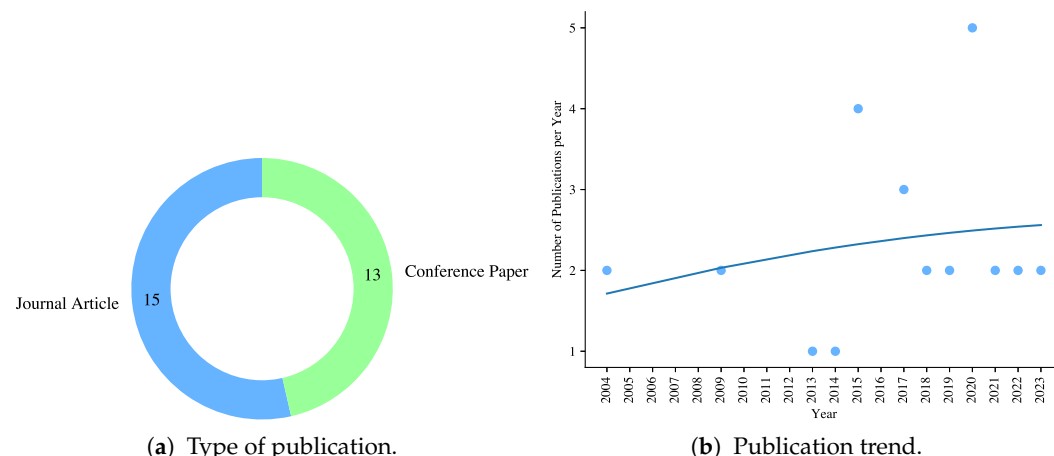

(**a**) Type of publication.

(**b**) Publication trend.

**Figure 3.** Shares of works published as journal articles and as conference papers (**a**) and historical trends of selected records (**b**).

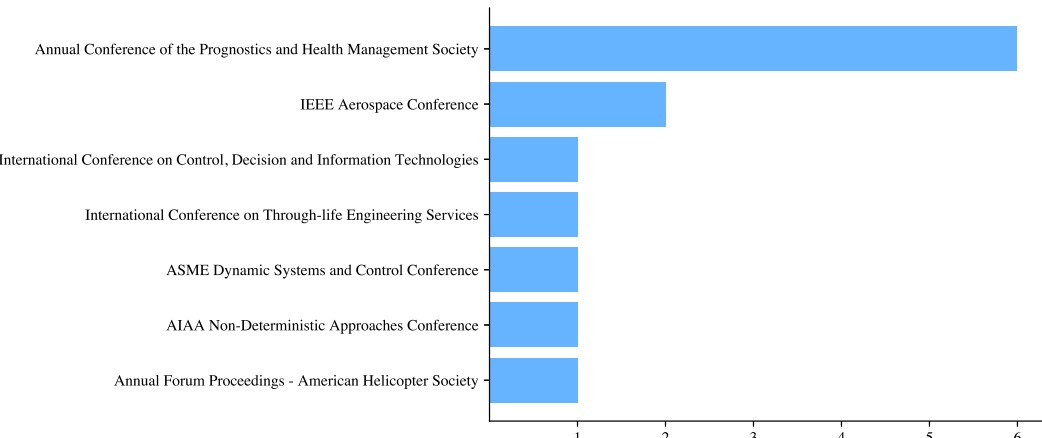

**Figure 4.** Most frequent conferences for the selected records.

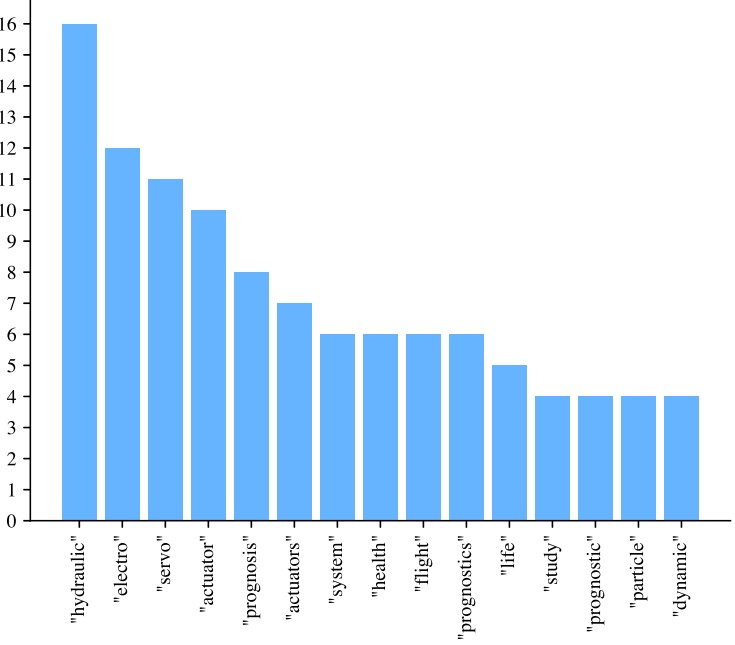

**Figure 5.** Words which appear more often in records titles as obtained through text processing.

## 3. Discussion

In this section, the research questions are analyzed and answered in detail. The answers are informed and driven by the material found in the selected records.

*3.1. RQ1: What Is the State of the Art of PHM in EHAs for Primary Flight Controls?*

With the growing interest towards PHM strategies, it may then seem trivial that FCS-related subsystems are gradually being more covered by these technologies. However, this is true only to some extent. While the buildup of constantly growing interest in the MEA concept has led many prognostic research activities related to EMAs, the hydraulic world has been a little bit left behind. In other words, there is an ongoing strong research effort in the direction of PHM systems for EMAs, but a limited amount of literature material involving PHM methodologies for EHAs is published in peer-reviewed journals. At the same time, literature reviews on PHM topics for EMAs can be found, but no analysis of the state of the art has been found for hydraulically powered solutions [4]. To put it in another way, literature on prognostics for EHAs is currently rather fragmented, scattered, and mostly focused on a few single fault scenarios or, on the other hand, mostly related to the analysis of one isolated component or only to the failure detection and identification without placing importance on the prognostic part. As a result, several articles focus on partial steps towards integrated PHM systems. The most relevant literature review on a similar topic is the one carried out by the researchers of the IVHM Centre, based in Cranfield, UK. In fact, the authors in [92] have performed a very recent review on diagnostic methods for hydraulically powered FCSs, comprising a summary of the main flight control actuator system configurations and methodologies without, however, focusing on the prognostic part. The causes of this imbalance between hydraulically and electrically powered actuation systems can be seen from different points of view. For instance, it is deemed that the future of aviation is going to be "more electric" or even "all electric" (All Electric Aircraft—AEA) [2]. The aforementioned MEA paradigm is slowly but surely changing the way mobile surfaces are actuated, and a trend towards the deployment of EMAs is in progress. Researchers are therefore focusing on future applications of EMAs [93]; however, the transition to widespread EMA adoption in civil aviation remains a long-term objective, requiring several more years of development and validation. If secondary flight controls are becoming a flourishing testing platform for electrically powered actuators, at the current state, primary flight controls are still powered by EHAs or EHSAs due to a series of challenges that span from thermal management issues and the need for inherent fault-tolerant architecture to mechanical complexity, wear, and power density. For instance, EMAs are more susceptible to single points of failure such as gear jamming or motor faults, hence often requiring complex redundancy schemes to meet safety standards. Another limiting factor is thermal management: under continuous load, EMAs generate significant heat, which is difficult to manage without dedicated cooling systems. Additionally, current EMA designs tend to have lower power density compared with hydraulic solutions, making them less suitable for high-force applications. Mechanical wear, backlash, and lubrication demands further complicate EMAs' life-time performance and maintenance. Finally, from a regulatory standpoint, EMAs lack the extensive operational history needed for certification. All these reasons, deeply analyzed in Maré [94], combined with the consequent lag behind in terms of certification and regulatory maturity, prevent a seamless introduction of EMAs as primary flight control actuators. In conclusion, if a shift towards EMAs is deemed plausible, for now, primary flight controls are, to all intents and purposes, still hydraulically controlled, and that is why this review focuses on this specific kind of actuator. Moreover, the development of data-based solutions is definitely easier for EMAs and electrically powered systems, given the extremely higher amount of digitalization and sensors. The

same can be said for aircraft turbofan engines, one of the most monitored subsystems on board. This explains the plethora of studies on PHM for aircraft turbofan engines and the availability of some datasets on this topic (e.g., NASA C-MAPSS [10]). Electro-hydraulic primary flight controls have always been characterized by a scarcity of sensors, and the signals used to close the control loop (e.g., spool position, jack position, and currents) are often not stored in any memory. On top of that, the idea of performing PHM checks on aircraft actuators had been conceived when EHAs were already proven technologies and, hence, industry and academic institutions prefer to focus on newer technologies, such as the electric ones. In other words, it is much more difficult to develop PHM solutions on already existing (and flying) equipment or legacy aircraft rather than designing solutions for the next generation of (maybe) electrically powered aircraft [95]. Moreover, industries that have developed and are developing PHM systems for hydraulic technologies do not disclose their solutions. Finally, the lack of precise and extensive data, as well as the major difficulties in understanding and modeling failure mechanisms, adds one more difficulty layer to an already demanding task [87], which, however, deserves attention and can prove to generate extensive savings [96].

### 3.2. RQ2: What Are the Most Prominent Authors, Affiliations, and Geographic Areas with the Highest Number of Records?

Figure 6a identifies the prominent authors in this field, namely the following:

- Jacazio Giovanni and Sorli Massimo from Politecnico di Torino, Italy.
- Guo Runxia from Civil Aviation University of China, China.
- De Martin Andrea from Politecnico di Torino, Italy.
- Vachtsevanos George from Georgia Institute of Technology, USA.

In particular, Jacazio Giovanni, Sorli Massimo, and De Martin Andrea frequently collaborated on their research. Figure 6b presents a pie chart diagram that illustrates the geographical distribution of the records, highlighting the prominence of affiliation from China, USA, and Italy. Figure 7a shows a bar graph of the specific affiliation of each author, underscoring the high presence of Politecnico di Torino, Italy with nine affiliations, Beihang University, China with four affiliations and Civil Aviation University of China, China with four affiliations. Finally, Figure 7b shows that most of the affiliations (30) are linked with the academic field and only five affiliations come from industrial institutions, namely: Impact Technologies (two records), Collins Aerospace (two records), Embraer (one record). The low number of contributions from industry may indicate barriers to the implementation of PHM solutions in real operational environments. It is also possible that more research is conducted within industry, but results are kept confidential due to competitive or proprietary constraints. This is particularly plausible in the aerospace sector, where safety-critical systems and IP protection restrict publication.

### 3.3. RQ3: Which Are the Most Used Approaches (Data-Driven, Model-Based, Hybrid) for Diagnosis and Prognosis?

As stated in the exclusion criteria, only articles with at least a prognosis part have been considered in this study. As a result of a thorough review of each record, the records have been categorized into three groups: those with both diagnosis and prognosis sections, those with both diagnosis and performance assessment sections, and those with either a prognosis section or a performance assessment section. The terms "prognosis" and "performance assessment" both involve analyzing the current state of a system and predicting its future health. However, specifically, "prognosis" refers to the Remaining Useful Life (RUL) estimation phase in the health assessment process, while "performance assessment" includes a generic health assessment analysis of the degraded system under

future operating conditions. Among the 28 records reviewed, 19 were classified as "diagnosis and prognosis", two as "diagnosis and performance assessment", five focused solely on "prognosis", and two introduced a "performance assessment" strategy, as shown in Figure 8. The complete and detailed list of the layers approached by each study is reported in Table A1 in Appendix A.

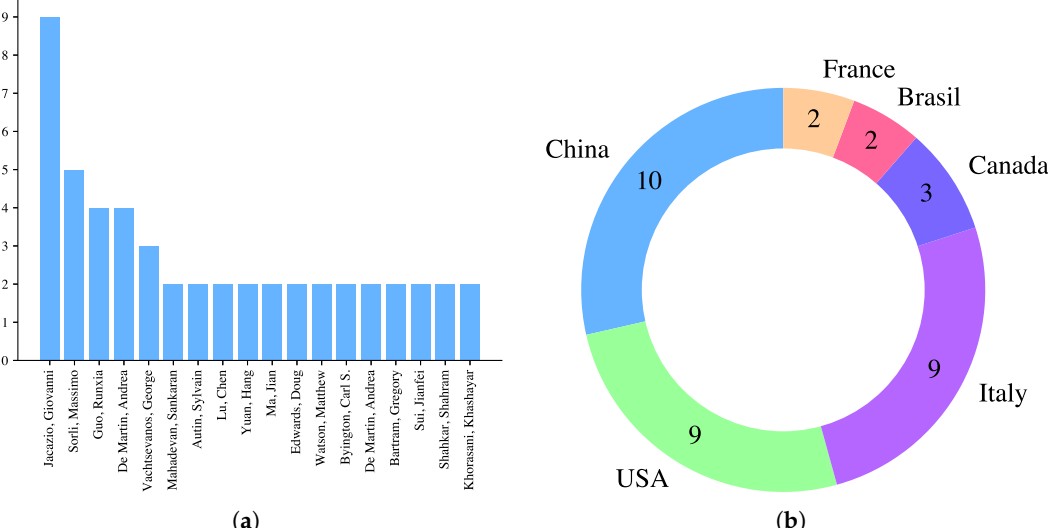

(a)　　　　　　　　　　(b)

**Figure 6.** Statistics of the most prominent authors (**a**) (authors who have published more than one article are reported) and the geographical area of the authors' affiliations (in articles where more than one authors were affiliated with the same institution, only one geographical record was taken into account) (**b**).

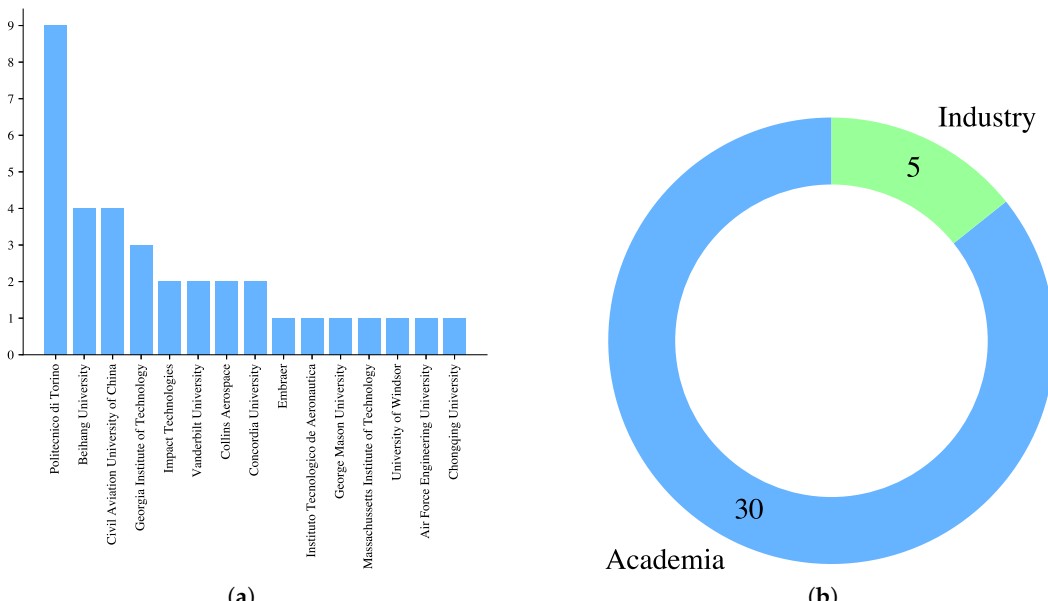

(a)　　　　　　　　　　(b)

**Figure 7.** Bar plot of the affiliations (in articles where more than one authors were affiliated with the same institution, only one affiliation was taken into account) (**a**) and pie chart of the affiliation type (**b**).

Each step has then been classified according to the employed approach: Data-Driven, Model-Based, and Hybrid. For the sake of clarity, in these charts, the Model-Based definition also includes Knowledge-Based and Physics-Based approaches. The results of this investigation are reported in Figure 9; in particular, Figure 9a refers to the diagnosis layer, Figure 9b is related to the prognosis layer, and Figure 9c highlights the approaches

for the overall strategy. "Diagnosis" (when included in the paper) is mostly approached with a Data-Driven strategy. This choice highlights the enormous amount of information contained in the data and underlines that, when recorded data are available, a Data-Driven diagnosis approach in the service is the most used. On the other hand, seven papers approach a Model-Based diagnosis strategy. Finally, seven papers do not include the diagnosis step and are hence highlighted in red. As far as "Prognosis/Performance Assessment" is concerned, the majority of the studies employ a Model-Based approach. This result is significant because it highlights that Model-Based strategies are used to contain and quantify uncertainty in the "prognosis/performance evaluation" phase. Data-Driven strategies are used in 32% of the studies (nine out of 28). The last pie plot considers the overall paper strategy: in this case, the situation is more uniform, and the Hybrid approach is the most used, underscoring that the mix between the Data-Driven and Model-Based approaches provides better results by combining the best of each method. Nine studies employ a Model-Based strategy only, and seven use a purely Data-Driven methodology. The complete and detailed list of the strategy type employed for each layer is reported in Table A2 in Appendix A.

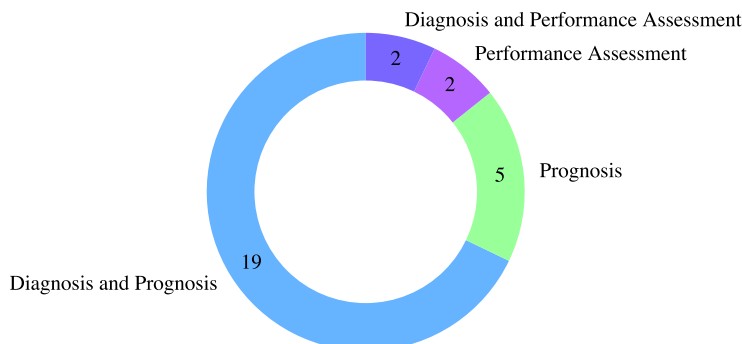

**Figure 8.** Approached functional layers: Diagnosis, Performance Assessment, and Prognosis.

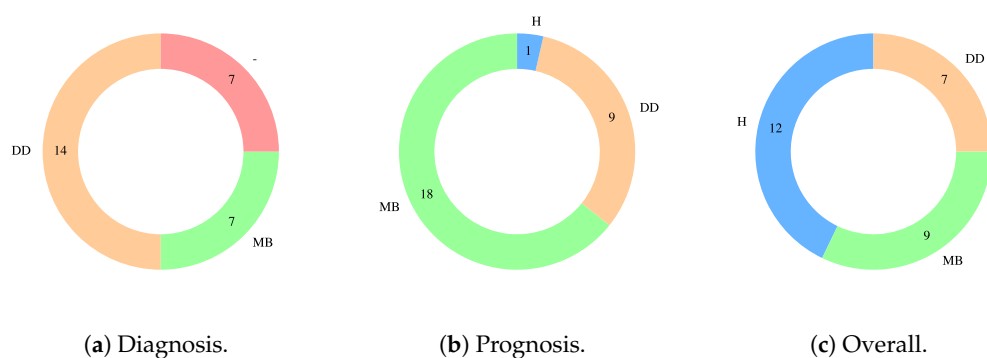

(**a**) Diagnosis.       (**b**) Prognosis.       (**c**) Overall.

**Figure 9.** Analysis of the approached methodologies (Data-Driven (DD), Model-Based (MB), and Hybrid (H)) in each phase of the PHM pipeline: Diagnosis (**a**), Prognosis (**b**), and Overall strategies (**c**).

Some final conclusions can be drawn from this analysis: the Data-Driven strategy is mostly employed in the "diagnosis" step, while the Model-Based strategy is adopted for the "prognosis/performance assessment" phase. For instance, an effective "Diagnosis" Data-Driven solution adopted by eight records is represented by straightforward signal distribution comparisons, which may highlight a divergence in one or more features. Of course, there is a need for representative datasets, and the signals need to be logged. In contrast, some Model-Based strategies, such as Dynamic Bayesian Networks (DBNs), can be employed when there is a need to integrate missing data. In the "Performance Assessment/Prognosis" phase, the use of Model-Based strategies is justified by the need to quantify the uncertainty and to project selected features into the future with an uncertainty

measure, as later explained in Section 3.6.1. This is why Particle Filter (PF) based strategies or similar Bayesian approaches are so frequently employed. As a result, the overall methodologies are Hybrid for the 42 % of the records. As a preliminary conclusion, a significant trend towards the application of platform-specific Data-Driven methodologies for the "diagnosis" step can be observed, while a more precise and generalizable Model-Based strategy is adopted for the majority of "prognosis/performance assessment" methods. Data-Driven strategies are only applied in 25 % of the overall strategies. This result confirms the trend towards the implementation of Hybrid methodologies and supports the overall research interest in the development of physics-informed strategies that can merge the specificity of Data-Driven solutions with the accuracy of Model-Based methodologies.

### 3.4. RQ4: Which Are the Most Investigated Components and Fault Modes?

EHAs are complex multi-disciplinary pieces of equipment encompassing a wide range of components that span multiple engineering disciplines (e.g., electrical, hydraulic, and materials science). As a result, several studies have approached the development of PHM strategies, often focusing on specific components rather than others. The analysis shown in Figure 10 provides a comprehensive overview of the most frequently investigated components from the selected records, shown in two graphical formats. Eleven studies concentrated on PHM related to hydraulic cylinders, while five focused on amplifiers and motors. Additionally, four studies examined the feedback spring and spool, and four others employed an agnostic methodology to develop strategies applicable to generic failures. Other components include hydraulic filters (three records), the overall mechanism associated with the actuator (three records), actuator structural integrity in relation to damage (three records), jet pipe systems (two records), seals (two records), and sensor deterioration (one record). One of EHAs' most complex and interdisciplinary components is undoubtedly the servo valve, which is the interface between the electric and hydraulic systems. Servo valves play a vital role in converting analog or digital input signals into precise and continuous hydraulic outputs, thus controlling hydraulic flow rate and/or pressure. In this sense, many of the already mentioned components are linked to the servo valve health status (e.g., amplifier, filter, jet pipe, etc.). There is a significant majority of research on hydraulic-related components, as opposed to electrical components, which appear to be investigated less frequently. This disparity may indicate that hydraulic faults are more easily identifiable and traceable, while the study of electrical components poses greater challenges due to a limited availability of sensors and apparent randomness in failures.

An additional analysis from a functional perspective has been conducted, categorizing the various fault modes in the deterioration process of the studied components (Figure 11). This analysis has identified a significant interest in specific research areas. Remarkably, 13 studies have focused on leakage in various components, such as cylinders, seals, and spools. Moreover, six studies worked on wear processes affecting structure, seals, and filters, while five studies investigated crack faults, clogging, and backlash in different components. Friction and motor degradation were approached in four studies each, while other minor fault modes were researched in fewer papers. The primary fault modes of EHAs are closely associated with leakage, and the high volume of studies addressing this issue reinforces its importance. Furthermore, leakage as a degradation process is closely related to wear and cracking of seals or other materials within the actuator body. As mentioned in the previous paragraph, electrical failure modes appear to be studied less frequently than their hydraulic counterparts. The complete and detailed list of the components investigated by each study, along with the fault modes, is reported in Table A3 in Appendix A.

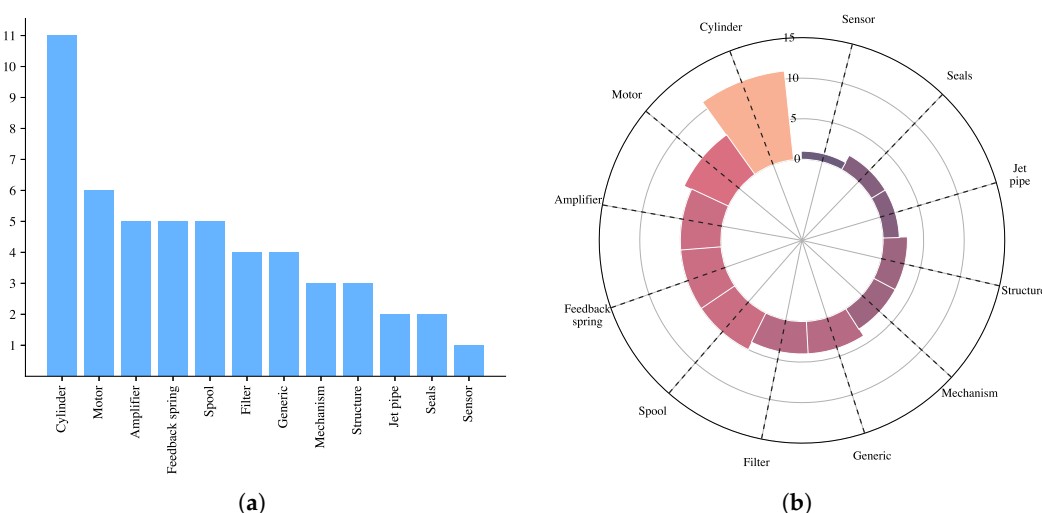

**Figure 10.** Visualization of the most frequently investigated components in the selected records: bar plot (**a**) and spider plot (**b**).

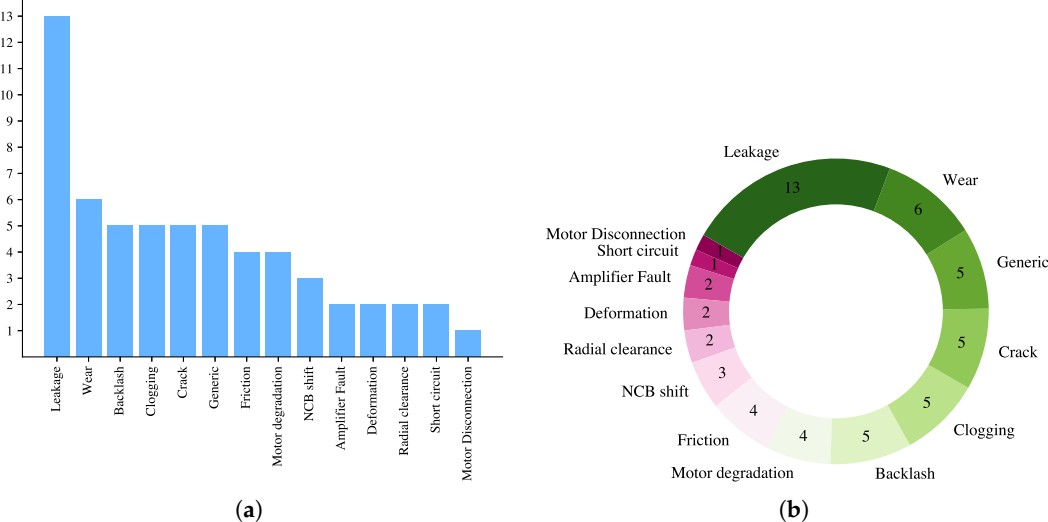

**Figure 11.** Visualization of the most frequently investigated degradation/fault mode: bar plot and pie chart: (**a**) Most frequently investigated degradation/fault modes: bar plot. (**b**) Most frequently investigated degradation/fault modes: pie chart.

### 3.5. RQ5: Which Are the Most Commonly Used Signals?

One of the primary challenges and initial analyses to undertake prior to conceptualizing a PHM system for operating equipment is the assessment of data and sensor availability. In fact, a fundamental step in the development of PHM strategies is the extraction and selection of informative and meaningful features that can describe and quantify the extent of the desired monitored faults. A feature set in this context can be defined as a reduced set of the available measurements that can be linked precisely to the health status of the system. In other words, features are informative signatures or fingerprints of a system selected or created starting from raw data to better represent the underlying problem. Or again, features are signals that contain high-value information, representative of the selected fault, correlated with the fault severity, and possibly not related to the presence of other failure modes [94,97]. Starting from a single logged value, an infinite number of features can be obtained by applying statistical functions (e.g., statistical moments like kurtosis, standard deviation, mean, median, etc. [98]), algebraic personalized functions, machine learning (ML) techniques, or Deep Learning (DL) algorithms. Principal Component Analysis (PCA),

Singular Value Decomposition (SVD), or the encoder part of AutoEncoders (AE) may come in handy to reduce the data dimensionality and select the signals with the higher information content [99,100]. These methodologies are usually employed to deal with high-dimensional feature space, but they can also be used to select or extract the right feature(s) in a smaller dimensional space. As the number of features can be very high, feature selection and extraction methods can be really helpful in identifying the most relevant features [100]. As also highlighted by the authors in [100], in some specific applications, "a smaller, less costly feature set with lower predictive ability might be preferable over a larger, more costly feature set with better predictive ability". In the case of EHAs, where the number of monitored signals is relatively limited, determining a representative value can prove to be quite complex and time-intensive. Furthermore, the approach may vary depending on whether the PHM engineer is asked to develop a PHM system from the ground up or if the equipment to be monitored is already in operation. In fact, in the former scenario, the designer can conceive a standalone architecture with tailored sampling rates, bit number, resolution, and sensor selection. From a design perspective, the integration of built-in test sensors within the overall system architecture facilitates continuous monitoring throughout the life cycle of the system or its components, thereby making the process substantially easier [92,100]. Conversely, in the latter scenario, engineers must utilize existing resources and frequently develop retrofits that are highly specific to individual platforms. Additionally, in the aerospace sector, where reliability, safety, weight, spatial constraints, and power consumption are paramount design considerations, the circumstances become increasingly critical as the integration of sensors is approached with greater caution [101]. In this case, in fact, the signals are often not always monitored, or they are logged at very low sampling rates or are difficult to access, especially if the product was designed some years ago, when the PHM revolution and the related applications were not even thought of. On the other hand, from an actuator OEM perspective, limiting the number of sensors is pivotal to increase the logistic reliability of the actuator and limit the complexity. Furthermore, the sensors used in a system are usually designed to work at the component or system level and, thus, may not be ideal to account for the interdependencies between subsystems at a higher level.

The typical sensor suite available in EHAs depends on the actuator type, its safety assessments and, sometimes more importantly, its age. Most legacy actuators only have two signals available: position information provided through one (or more) LVDTs and the servo valve currents signal, which corresponds to the output of the control laws. This kind of setup clearly limits the extent of the fault mode that can be observed and the type of PHM analysis that can be pursued. More recent actuators can sometimes rely on the position information of the servo valve spool (for control or to monitor the possible occurrence of runaways), and of the pressure differential between the actuator's chambers (mostly for monitoring purposes, in some instances for control). A richer setup can significantly improve the performance of any diagnostic/prognostic scheme, as it allows for more easily separating faults affecting the servo valve and to better identify at any time instant the operational scenario faced by the actuator (extraction/retraction, load type, etc.). A critical point to discuss on this topic pertains to the cut-off frequency typically employed in such sensors; as they are typically defined for control purposes, the measured signals are typically associated with a bandwidth in order to remove what is traditionally perceived as high-frequency noise. However, data collected at these frequencies may contain significant information relevant to PHM applications.

Given the importance of signals and features, the following paragraph provides a comprehensive overview of the types of sensors and signals commonly utilized in the development of PHM frameworks for EHAs in the selected records. This analysis is

extremely useful in designing a strategic approach that is both forward-looking and well-founded. The findings indicate that five main signals are routinely exploited and appear more than ten times in the selected records: actuator position (19 records), differential pressure (11 records), spool position (11 records), command (10 records), and servo valve current (10 records). Although other signals are monitored, they are used less frequently within PHM frameworks; for example, oil temperature is noted in three records, and internal valve position and/or current are documented in three and two records, respectively. A broader array of additional signals is employed only occasionally for specific fault detection, as illustrated in Figure 12.

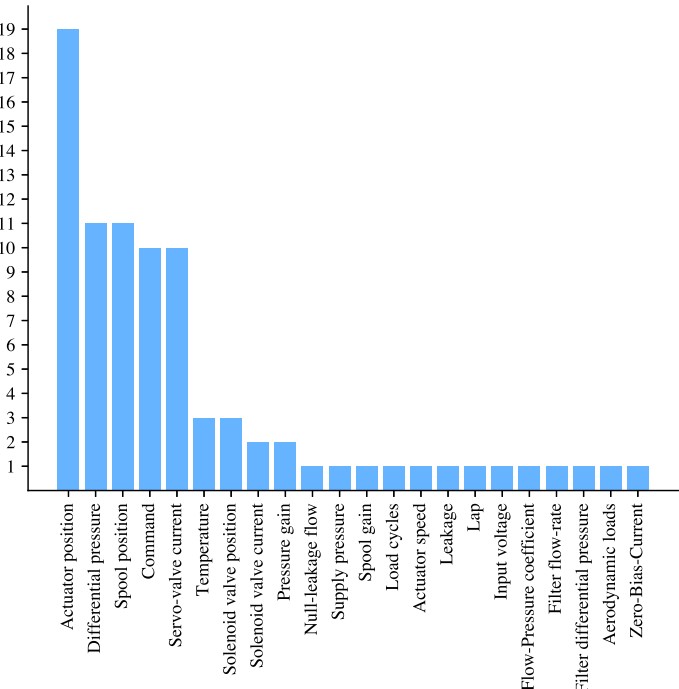

**Figure 12.** Bar plot of the most used EHA signals.

The actuator position is utilized in 19 studies, making it the most informative signal to log for monitoring the health status of EHAs. This parameter offers valuable insights into the presence of external faults and provides actionable data regarding the system dynamics, such as frequency response and resolution tests. Additionally, it is easily measurable externally with a transducer, making it an optimal choice and an essential component for analysis. In addition, differential pressure and spool positions are included in a total of 11 studies. However, these signals are more challenging to log unless a specific sensor has been installed for that purpose. On the other hand, the internal actuator signals can provide significant advantages, as they facilitate the identification of internal leakages and deterioration processes. Bertolino et al. [87] selected as the feature for EHA internal leakage the ratio between the RMS of the spool displacement and of the pressure drop, computed on a moving window of 1 s. Another example is the one reported in Dalla Vedova et al. [102], where the authors identified the classical dynamics characteristics of the spool position (i.e., delay time, rise time, settling time, peak overshoot, time to peak) and the maximum spool position and actuation speed as potential features for a Model-Based FDI strategy exploiting ANNs. The command and servo-valve current are employed to correlate input and output variables, allowing for comprehensive monitoring of the system's internal behavior. For instance, the authors in [65] used the Fast Fourier Transform (FFT) of the output valve pressure, a frequency analysis of the servo current, and a control valve position feature obtained through a feed-forward, time-delay NN error tracking

modeling (servo-current, commanded ram position change, and feedback valve position are inputs to the NN). In conclusion, a total of five signals are mostly used across the studies (i.e., actuator position, differential pressure, spool position, command, and servo valve current), establishing a valuable starting point for the development of EHA PHM systems.

Another important consideration involves the signal sampling rate, which is essential for monitoring specific behaviors. For instance, the authors in [90] used a 6 kHz signal of the Zero Bias Current (ZBC) of an Integrated Servo Actuator (ISA). Liu et al. [70], logged the output signal with a sample frequency of 1 kHz with a sinusoidal input signal and a total sampling time of 6 s. The work carried out in [103] suggests time series data of at least 25 Hz for the aircraft dynamics. Autin et al. [78,86] state that the high-fidelity model used in their PHM framework, implemented in Matlab (Simulink), runs at a fixed integration step of $10^{-4}$ s and a 2 Hz sinusoidal command is used as the source signal. A PHM scheme for a multiple redundancy aileron actuator (MRAA) is presented in [77]. In this case, the simulation time of these tests (with a model developed in Matlab Simulink and AMESim) was 240 s, and the sampling rate was 10 samples/s. Of course, the characteristics of the monitored phenomenon are critical: in cases where the dynamics are rapid, such as with electronic signals, a high sampling rate is required to capture all relevant behaviors. Furthermore, it is common to encounter multi-rate data sampling, which exhibits issues such as inadequacy (missing values in low-sampled data compared with high-sampled data), consistency, and information asymmetry. Typically, in a functional process, high-sampling-rate variables are predominantly process variables that provide limited process information, whereas low-sampling-rate variables tend to be more quality-related, thereby containing more significant information content [92]. On the other hand, PHM engineers have to consider the energy and sustainability of the chosen sampling rate [100]. Some recent studies show the possibility of signal up-scaling and up-sampling using Deep Learning techniques, starting from a very in-depth knowledge of the system being monitored [104]. These techniques may come in handy to synchronize multi-rate data samples, but there is a need to quantify and certify the uncertainty added to the signal. The complete and detailed list of the signals used by each study is reported in Table A4 in Appendix A.

### 3.6. RQ6: Which Methods and Techniques Are the Most Used Ones?

After the analysis of the primary trends within the functional layers (i.e., diagnosis, prognosis, and performance assessment), the methodologies employed (i.e., Data-Driven, Model-Based, or Hybrid approaches), the most frequently investigated components, the predominant degradation or fault modes, and the most utilized signals in the previous section, a detailed description of the selected records is presented. The primary objective is to emphasize the similarities and differences among the selected approaches, as well as to discover patterns and possible unexplored research opportunities. After a careful review of each record, Tables 2 and 3 report the employed methodology used to carry out diagnosis and Performance Assessment/Prognosis in each study.

**Table 2.** Diagnosis method adopted in each study.

| Ref. No. | Diagnosis |
|---|---|
| [64] | Fuzzy logic classifier on three features, FFT on hydraulic pressure, Electric current Signature Analysis (ESA) on the servo valve current, and a feed-forward neural network |
| [65] | Fuzzy logic classifier on three features: FFT on hydraulic pressure, Electric current Signature Analysis (ESA) on the servo valve current, and a feed-forward neural network |

**Table 2.** *Cont.*

| Ref. No. | Diagnosis |
| --- | --- |
| [66] | Residue approach on feature integral with threshold chosen via frequency responses |
| [67] | Custom mathematical functions applied on signals and thresholds |
| [68] | System model based on Dynamic Bayesian Network, Particle Filter |
| [69] | - |
| [70] | Elman neural network observer, Gaussian Mixture Model (GMM) |
| [71] | System model based on Dynamic Bayesian Network, Particle Filter |
| [72] | Data-Driven distribution comparison |
| [73] | Mahalanobis distance applied on features obtained through a Mean Impact Value guided optimization on Radial Basis Function (RBF) neural network state observer obtained error |
| [74] | Two-step identification, Matrix Regressor Adaptive Observers (MRAO) |
| [75] | - |
| [76] | Data-Drive distribution comparison |
| [77] | Two-step RBF neural network (observer and error computing) |
| [78] | Data-Driven distribution comparison; Non linear symbolic regression |
| [79] | Data distribution comparison, Modeled features |
| [80] | - |
| [81] | - |
| [82] | Three distributed Multi-Layer Perceptrons (MLPs) |
| [83] | - |
| [84] | Data-Driven distribution comparison |
| [85] | Data-Driven distribution comparison |
| [86] | Data-Driven distribution comparison |
| [87] | Data-Driven distribution comparison, Linear SVM |
| [88] | Data-Driven distribution comparison, Linear SVM |
| [89] | Multidimensional Bayesian methodology |
| [90] | - |
| [91] | - |

**Table 3.** Performance Assessment/Prognosis method adopted in each study.

| Ref. No. | Performance Assessment/Prognosis |
| --- | --- |
| [64] | Feature-based state space tracking routine (Kalman filter) with Newtonian relationship |
| [65] | Feature-based state space tracking routine (Kalman filter) with Newtonian relationship |
| [66] | RUL linear interpolation |
| [67] | Threshold-based system on the absolute position error: Least square interpolating function or linear projection depending on the fault level |
| [68] | System model based on Dynamic Bayesian Network and Sequential or Recursive Monte Carlo (Particle Filter) |
| [69] | Physics of Failure (PoF), mathematical models for wear |
| [70] | Support Vector Regression (SVR) |
| [71] | System model based on Dynamic Bayesian Network and Sequential or Recursive Monte Carlo (Particle Filter) |
| [72] | Particle Filter, High-fidelity model |
| [73] | Elman neural network |
| [74] | Graph extrapolation on a feature map graph |
| [75] | F-Distribution Particle Filter |
| [76] | Particle Filter, High-fidelity model |
| [77] | Self Organizing Maps (SOM) |
| [78] | Particle Filter, High-fidelity model |
| [79] | Particle Filter |

**Table 3.** *Cont.*

| Ref. No. | Performance Assessment/Prognosis |
|---|---|
| [80] | Support Vector Regression (SVR) and Particle Filter based on Kendall correlation coefficient |
| [81] | Minimum Hellinger Distance on a Particle Filtering (PF) algorithm |
| [82] | Recursive Bayesian algorithm |
| [83] | Improved relevance vector machine |
| [84] | Particle Filter, High-fidelity model |
| [85] | Particle Filter, High-fidelity model |
| [86] | Particle Filter, High-fidelity model |
| [87] | Particle Filter, High-fidelity model |
| [88] | Particle Filter, High-fidelity model |
| [89] | Bayesian multidimensional space methodology |
| [90] | Nonlinear Wiener Process (NWP) and Wavelet Packet Decomposition Echo-State-Network (WPD-ESN) |
| [91] | Exponential Smoothing, ARIMA, and fusion prediction |

3.6.1. The Need for Uncertainty Assessment and Bayesian Algorithms

A notable pattern is the necessity to assign an uncertainty metric in conjunction with the predictive assessment of the system's health status. This requirement is inherent in the definition of PHM, which suggests an uncertainty assessment in order to provide decision-makers with a definitive and justifiable knowledge of asset health [97]. This trend is evident in the widespread application of Bayesian statistics, utilized both in the diagnostic phase and, more prominently, in the prognostic phase. For instance, a DBN is used in two papers by Bartram and Mahadevan [68,71], while a multidimensional Bayesian methodology is used in [89]. The DBN employed in [68,71] is used for two main reasons. On the one hand, it enables the integration of different sources of information: expert insights, reliability data, various mathematical models (including system state space models and physics of failure models), established databases of operational and laboratory data, as well as real-time measurement information. On the other hand, a DBN is selected because it can handle uncertainty in diagnosis that is then propagated forward in the prognosis step. In line with this approach, one Bayesian algorithm that is widely utilized is the PF [105–107]. PFs have been employed in as many as 13 prognosis approaches, making them one of the most used tools in the field. PFs are probabilistic failure prognostic algorithms that estimate the future health state of the system based on a degradation model and measurements. In particular, the algorithm monitors and predicts the state of health of the system by tracing the probability density function of a number of weighted samples, or "particles," evolving in time. The PF owes its widespread success to the possibility of including the notion of uncertainty in the process, making it an optimal choice to provide estimation with the level of uncertainty and for its capability in handling nonlinear and non-Gaussian distributed data. PFs present a long history of applications in engineering, and they have been used extensively in the PHM field for the aforementioned reasons. During the years, some modifications have been applied to solve some inefficiencies (e.g., resampling techniques). In this sense, the PF presented in [81] is modified by adding a Minimum Hellinger Distance evaluation to cope with and solve the particle degeneracy problem at the early phase of the prediction, where most weights are focused on a few samples. Guo and Gan in [75] proposed a way to improve the PF performance by modifying the way particle weights are updated and introducing a methodology combining the F-distribution with traditional PF to dynamically predict the future state. Guo and Sui in [80] showed another variation using the Kendall correlation coefficient to improve the particle degeneracy problem combined with Support Vector Regression (SVR) results, which are

used as inputs for the PF. The Bayesian multidimensional approach presented in [89] is said to show advantages over traditional recursive numerical methods (e.g., PF), particularly in its ability to utilize specific information, such as sensor readings, for optimal inference. In contrast, conventional recursive algorithms frequently rely on aggregating information to manage the dimensionality of the variables.

### 3.6.2. AI Implementation and the Necessity of Explainability and Robustness

The use and reliance on AI strategy is limited and is mostly used only in the diagnosis step to obtain features. This choice may be traced back to the need for an explainable and trustworthy prognosis algorithm and to the requirement to provide maintenance decision-makers with results that can be traced back to physical signals and data. For example, a simple feedforward neural network has been employed in [64,65] to calculate one of the three features used to monitor and detect generic faults in EHAs. A DBN has been used in [68,71] to try to combine the customization capability of AI solutions and the robustness and uncertainty handling capability of Bayesian statistics. A Radial Basis Function (RBF) neural network has been used in [59,73], using RBFs as activation functions, leveraging their nonlinear mapping capabilities and training efficiency. In particular, in the diagnostic phase in [77], a two-step neural network is used, consisting of two RBF neural networks. The first network is responsible for monitoring the MRAA and generating the residual error, while the second network simultaneously produces the corresponding adaptive threshold. Three distributed Multi-Layer Perceptrons (MLPs) have been used by Kordestani et al. in [82] in a custom-built failure parameter estimation unit to monitor three different fault modes: actuator leakage coefficient degradation, null bias current shift, and internal leakage. Only one MLP branch is active while monitoring one of the three specific faults by taking as input the data of the control feedback signal and the output position from an LVDT sensor at each sampling time. The goal of the distributed network is to estimate the real failure parameters in time. The powerful dimensionality reduction capabilities presented by self-organizing maps (SOMs) have been used in [77] to assess actuator performance with the added benefit of maintaining the topological properties of the input feature space unchanged. An Echo State Network (ESN) has been used by Cui et al. [90] combined with a Nonlinear Wiener Process (NWP) to generate more degradation data. In particular, the capability of handling time series data of the Recurrent Neural Network (RNN)-based ESN algorithm is utilized to characterize the physical degradation process of ISA. Simpler ML algorithms such as linear Support Vector Machines (SVM) trained on a simulated dataset and verified through a k-fold cross-validation process are employed in the diagnostic phase to classify the faults by the authors in [87,88]. A more generalized version of the SVMs are SVRs, employed in the studies [70,80] for the prognostic phase.

### 3.6.3. Observers and Simplicity

A notable trend involves the use of observers, systems that deliver an estimate of the internal state of a specified real system based on measurements obtained from the input and output of the system [108]. The authors in [70,73,74] all employed neural network-based or Matrix Regressor Adaptive Observers (MRAO) to obtain hidden states of the actuator system. Lu et al. [77] used an RBF neural network-based observer to estimate the output of the actuator system. Finally, it has to be noted that many studies prefer explainable and relatively simple methodologies to approach diagnosis and prognosis steps; this is the case of simple feature data distribution comparisons used to detect and identify a possible fault. This is the strategy employed by the authors in [72,78,79,84–87,89], in which histograms of the features are compared and the dissimilarity between a nominal distribution and the actual distribution is assessed via different metrics. This is also the case

of the residue approach on feature integral and the RUL linear interpolation employed in De Oliveira Bizzarria and Yoneyama [66] to monitor the clogging of the actuator hydraulic filter, the usage of a feature map graph in Soudbakhsh and Annaswamy [74] applied on features derived from the flow-pressure coefficient and spool gain to monitor generic actuator faults, and the solution applied in Borello et al. [67] with simple mathematical expressions to calculate features and a least square interpolating function for the RUL estimation. The emphasis on simplicity is certainly noteworthy and should be regarded as an added value in a PHM solution, as it likely results in a more explainable and practically implementable system in operational scenarios.

### 3.7. RQ7: What Are the Current Challenges That Prevent PHM Solutions for Primary Flight Controls from Increasing the Product Availability and Cost-Effectiveness?

As stated in Section 3.1, the development of PHM strategies for EHAs remains limited and is constrained by various factors, which can be broadly categorized into technical and organizational challenges.

#### 3.7.1. Technical Challenges
#### Actuator System Knowledge and Degradation Models

EHAs are complex piece of machinery where different fields of engineering meet, and, as such, the multidomain system knowledge required to discern and understand the multiple failure modes is extensive. Many of the required degradation processes have not yet been investigated, nor is the effect of external factors (e.g., vibrations) or the combination of different factors acting together on the actuator during operations. Degradation models for complex systems like EHAs are another critical topic that prevents the implementation of high-fidelity models, which, on the other hand, can be developed more easily for different kinds of systems (e.g., disk brakes, batteries, etc.).

#### Few-Shots Phenomenon and Data Imbalance

Another important issue already explained is the so-called "few-shots" phenomenon, related to data imbalance. Airplanes are considered the safest means of transportation thanks to the countless safety assessments during the whole product life cycle, starting from the design phase when different Design Assurance Levels (DALs) are assigned to each part of the project, passing through the production phase, and all the way to the operational step, where maintenance is carried out according to strict and conservative criteria. Flight controls (especially primary flight controls) are safety-critical assemblies and, as such, require compliance with the strictest safety criteria. All efforts in the direction of reducing faults, failures, and accidents have led to the actuator failures being (fortunately) rare. What this means is that we can talk about a "few-shots" phenomenon that leads PHM engineers to very few operational faulty data. As already stated, this problem is more commonly known by the name of data imbalance, highlighting that there are many more healthy data points than faulty ones. This represents one of the main challenges PHM engineers face, as it hinders the training of traditional Data-Driven models (e.g., neural networks, XGBoost), making it difficult to ensure robustness across platforms or mission profiles, validate generalization without overfitting, and ultimately deliver statistically sound solutions. Over the years, several solutions have been proposed, although the problem is still far from having a universally recognized answer. In fact, the proposed strategies are often tailored to the specific degradation pattern or the equipment under investigation. For instance, the "few-shots" phenomenon has motivated research in the fields of transfer learning, Hybrid modeling, physics-informed techniques, and generative neural networks (e.g., Generative Adversarial Networks (GANs)) [109]. Transfer learning is a promising research direction that can merge research lines in the context of labora-

tory tests and real-world conditions. The idea of integrating physics knowledge in ML solutions, either via mathematical modeling or via Physics-Informed Neural Networks (PINNs), is providing a great leap forward in algorithm robustness but requires a sound knowledge and formalization of the degradation patterns under test. Finally, GANs can synthetically generate new data samples that resemble the limited real data, effectively augmenting the dataset [110,111]. However, the reliability of GAN-generated data in capturing realistic degradation behaviors must be thoroughly tested and validated under varying operational conditions.

Data Availability, Quality, and Intellectual Property Rights

Another open point common to the general PHM community involves data availability, data quality, and Intellectual Property Rights (IPRs). With the advent of Industry 4.0, it is often said that "data is the new oil" and this is particularly true for the PHM and CBM sectors. Data are difficult to acquire, and simulated, modeled, and synthetic data need proper (and costly) validation [92]. If a PHM framework for legacy equipment is considered, data are often scarce or not consistent with the basic notion of data quality (accuracy, completeness, consistency, and currentness) [112–116]. In addition, data are often not shared by industrial partners, as they are considered a strategic asset. Data are scattered among many actors in the field and need to be shared (between OEMs, airlines, and MROs). The problem of data closure in the industrial sector is often the blocking factor that does not allow a seamless use of existing datasets. As highlighted by the ReMAP (Real-Time Condition-Based Maintenance for Adaptive Aircraft Maintenance Planning) project, stakeholders can collaborate using a distributed IT approach to handle data in a trustworthy IT environment. Federated learning could indeed be employed to train models while safeguarding the confidentiality of company data [117]. Finally, the use of synthetic data generation through modeling could be used to fill the gap left by the lack of data, as already discussed in the previous paragraphs.

Knowledge Integration

PHM systems output must be probabilistic in nature and must carry information on the risk associated with taking decisions based on it in order to let decision makers make conscious and mathematically backed-up decisions. This kind of information is often very difficult to quantify but, on the other hand, is essential to enable a PHM system to provide actionable data that can be integrated in the overall management decision process. Another difficult task is seamless data integration and fusion from different data sources [5]. As such, the integration of Model-Based and Data-Driven approaches considering prior knowledge, information, and data are particularly difficult with operational and real-life data. Moreover, when a PHM system is developed for a legacy asset, data coming from different sources and formats must be integrated into a single coherent data management system in order to develop a forward-looking and efficient PHM framework.

Lack of Objective and Universally Recognized Evaluation Metrics

There is an ongoing effort in the PHM community towards the adoption of standard PHM metrics that can be employed to objectively evaluate, compare, and choose a strategy over another. In fact, due to the extensive variety of applications, scopes, data quality, data sources, and domains, a wide range of metrics and performance evaluations have been used, making the comparison of different approaches difficult and time-consuming. The authors in [118–122] have proposed a set of common and standardized PHM metrics that can be employed to make the comparison between strategies easier and more effective, nevertheless, extensive work remains to be conducted.

### 3.7.2. Organizational and Business Challenges

EMAs

As already mentioned in Section 3.1, the advent of EMAs is gradually shifting the research efforts of the flight control PHM community towards the study and development of PHM strategies and solutions for EMAs. This shift is driven by the expectation that, in the future, More-Electric Aircraft architectures will favor the replacement of EHAs with EMAs, as the latter offer greater compatibility with digitalization and sensing, thereby facilitating the development of PHM strategies. As a result, both research interest and business investments are increasingly directed toward PHM strategies for EMAs, which are expected to play a transformative role in the coming decades. This shift makes it more challenging to demonstrate the benefits of applying PHM to legacy systems such as EHAs, especially when the research focus is increasingly aligned with the more innovative EMA paradigm.

Industries Organization

In order to develop an integrated PHM strategy, there is an extreme need for data coming from every industry department: quality, engineering, customer support, MRO, etc. This horizontal level knowledge that crosses the organizational charts is not well integrated with the vertical and strictly siloed organization of engineering industries [123,124]. Additionally, as already mentioned in the previous paragraphs, data are key to developing grounded PHM systems. Stakeholders have to collaborate to develop trustworthy ecosystems in which data could be shared, as well as standards to safeguard IPRs and confidential information. Finally, the actionable asset information obtained at the end of the PHM strategy has to be integrated into an industry operational organization, which must be ready to trust, valorize, and exploit the obtained information. This mission must be integrated into broader, high-level digital transformation initiatives that permeate the organization from top to bottom, driving a fundamental shift in how data are handled, stored, and processed, and creating further incentives for the development of Data-Driven solutions. Data silos should be eliminated to facilitate a unified data stream that can track the asset throughout its life cycle, thereby supporting operational improvements.

Demonstration of an Acceptable Return on Investment

A recurring challenge in the development and deployment of PHM systems is the ability to clearly demonstrate their value and secure management approval prior to the beginning of the project. This issue is particularly pronounced in complex operational environments such as those that involve EHAs [95,125–127]. In the aerospace sector, where profit margins are typically narrow and operational risk tolerance is low, investments must be justified through tangible, quantifiable benefits. However, PHM benefits are often probabilistic or long-term, making them difficult to frame within traditional cost–benefit analyses. This creates a barrier to implementation, especially when competing with other investment priorities. Addressing this challenge requires the development of rigorous Return On Investment (ROI) assessment methodologies, tailored business cases, and validation through field data or simulations to effectively communicate the economic and operational advantages of PHM systems.

## 4. Conclusions and Future Directions

The SLR performed in this paper offers a structured and comprehensive overview of PHM strategies for flight controls powered with EHAs, highlighting recent advancements as well as the challenges encountered during their development. While the total number of relevant articles found and analyzed may appear limited, this reflects the actual availability

of focused studies on this specific topic, as rigorously determined by the adopted review protocol. As such, the findings presented here are not intended to serve as a definitive foundation for all future PHM developments in aerospace, but rather as a necessary and methodologically sound first step to guide further investigation in this critical domain. Particular care has been taken to make the review process as traceable, explainable, and objective as possible with the adoption of PICOT criteria, PRISMA workflow, and commonly used practices. The state of the art of PHM methodologies for EHA-powered primary flight controls has been extensively analyzed, highlighting the need for literature on this topic and underscoring the motivation of this work. The review highlighted some common points in the development of PHM strategies for EHAs, such as the reliance on Data-Driven methods for diagnosis and on Model-Based approaches for prognosis, where fault and degradation data are missing. Moreover, since feature selection is an essential step for the development of a robust and effective PHM system, a thorough and detailed discussion has focused on the different features selected in the most relevant papers found and highlighted the prevalence of a set of signals: actuator position, differential pressure, spool position, command and servo valve current. The performed analysis could be of paramount importance to PHM engineers, providing a grounded analysis to back up the development of monitoring strategies on existing/or brand-new systems. Additionally, the SLR identified the most investigated components and fault modes. The most investigated failure mode was leakage along with the hydraulic cylinder as the most researched component, underscoring the importance of a multi-disciplinary approach and proving once again the central role of degradation models. As discussed in Section 3.7, data represent one of the main bottlenecks in PHM strategy development. Based on the SLR results, it is deemed that possible future research directions of EHA PHM strategies should definitely concentrate on the enhancement of data quality and focus on the collaborative creation of shared degradation databases via research projects, which could merge the academic and industrial backgrounds. In fact, the observed imbalance reported in the context of RQ2 underlines the need for closer collaboration between academic researchers and industrial stakeholders. Joint research initiatives or public–private partnerships could foster more applied PHM solutions and accelerate the path toward industrial readiness. On top of that, the analysis proved once again the superiority of Hybrid approaches merging the data and models in a single framework. On the other hand, the use of synthetic data modeling has to be backed up by clean and reliable data obtained through real-life testing or test benches. Moreover, the integration of PHM routines into a more layered data architecture could definitely contribute to the concept of digital twins, which, with their two-way connection between a physical asset and a mathematical model, could improve the customized prediction ability of unscheduled removals. Furthermore, future PHM strategies for EHAs can take inspiration from the most recent advancements aimed at EMAs and EHSAs. These applications, despite being physically different, can be helpful in discovering novel diagnostic and prognostic techniques. This is particularly true for EHSAs, which, despite presenting a different electrical interface, share the same hydraulic working principles inside the EHSA itself. In conclusion, this SLR is a complete and useful reference providing a bird's-eye perspective of the complex panorama of PHM strategies for EHAs, starting from the very essential preliminary steps until the development of effective PHM solutions, which can then be included in maintenance optimization frameworks to improve the system availability and readiness.

**Author Contributions:** Conceptualization, L.B., A.D.M. and G.J.; methodology, L.B. and A.D.M.; software, L.B.; validation, L.B.; formal analysis, L.B.; investigation, L.B.; resources, L.B.; data curation, L.B.; writing—original draft preparation, L.B.; writing—review and editing, L.B. and A.D.M.; visualization, L.B.; supervision, G.J. and M.S.; project administration, G.J. and M.S.; funding acquisition, M.S. All authors have read and agreed to the published version of the manuscript.

**Funding:** This publication is part of the project PNRR-NGEU, which received funding from the MUR—DM 352/2022. This research is co-funded by Leonardo SpA.

**Data Availability Statement:** Data availability are not applicable to this article as no new data were created or analyzed in this study.

**Conflicts of Interest:** Leonardo Baldo reports financial support was provided by Leonardo Aircraft Division. If there are other authors, they declare that they have no known competing financial interests or personal relationships that could have appeared to influence the work reported in this paper.

## Appendix A

**Table A1.** Approached PHM layers for each record.

| Ref. No. | Approached PHM Layer(s) |
|---|---|
| [64] | Diagnosis, Prognosis |
| [65] | Diagnosis, Prognosis |
| [66] | Diagnosis, Prognosis |
| [67] | Diagnosis, Prognosis |
| [68] | Diagnosis, Prognosis |
| [69] | PerfAss |
| [70] | Diagnosis, PerfAss |
| [71] | Diagnosis, Prognosis |
| [72] | Diagnosis, Prognosis |
| [73] | Diagnosis, Prognosis |
| [74] | Diagnosis, Prognosis |
| [75] | Prognosis |
| [76] | Diagnosis, Prognosis |
| [77] | Diagnosis, PerfAss |
| [78] | Diagnosis, Prognosis |
| [79] | Diagnosis, Prognosis |
| [80] | Prognosis |
| [81] | Prognosis |
| [82] | Diagnosis, Prognosis |
| [83] | Prognosis |
| [84] | Diagnosis, Prognosis |
| [85] | Diagnosis, Prognosis |
| [86] | Diagnosis, Prognosis |
| [87] | Diagnosis, Prognosis |
| [88] | Diagnosis, Prognosis |
| [89] | Diagnosis, Prognosis |
| [90] | Prognosis |
| [91] | PerfAss |

**Table A2.** Type of adopted strategy for each phase, as well as for the overall strategy.

| Ref. No. | Strategy DS | Strategy PS | Overall Strategy |
|---|---|---|---|
| [64] | Data-Driven | Data-Driven | Data-Driven |
| [65] | Data-Driven | Data-Driven | Data-Driven |
| [66] | Model-Based | Model-Based | Model-Based |
| [67] | Model-Based | Data-Driven | Hybrid |

**Table A2.** *Cont.*

| Ref. No. | Strategy DS | Strategy PS | Overall Strategy |
|---|---|---|---|
| [68] | Model-Based | Model-Based | Model-Based |
| [69] | - | Model-Based | Model-Based |
| [70] | Data-Driven | Data-Driven | Data-Driven |
| [71] | Model-Based | Model-Based | Model-Based |
| [72] | Data-Driven | Model-Based | Hybrid |
| [73] | Data-Driven | Data-Driven | Data-Driven |
| [74] | Model-Based | Data-Driven | Hybrid |
| [75] | - | Model-Based | Model-Based |
| [76] | Data-Driven | Model-Based | Hybrid |
| [77] | Data-Driven | Data-Driven | Data-Driven |
| [78] | Data-Driven | Model-Based | Hybrid |
| [79] | Model-Based | Model-Based | Model-Based |
| [80] | - | Model-Based | Model-Based |
| [81] | - | Model-Based | Model-Based |
| [82] | Data-Driven | Model-Based | Hybrid |
| [83] | - | Data-Driven | Data-Driven |
| [84] | Data-Driven | Model-Based | Hybrid |
| [85] | Data-Driven | Model-Based | Hybrid |
| [86] | Data-Driven | Model-Based | Hybrid |
| [87] | Data-Driven | Model-Based | Hybrid |
| [88] | Data-Driven | Model-Based | Hybrid |
| [89] | Model-Based | Model-Based | Model-Based |
| [90] | - | Hybrid | Hybrid |
| [91] | - | Data-Driven | Data-Driven |

**Table A3.** Investigated components (focus area) and fault modes for each record.

| Ref. No. | Focus Area | Faults (Functions) |
|---|---|---|
| [64] | Generic | Generic |
| [65] | Generic | Generic |
| [66] | Filter | Clogging |
| [67] | Filter, Motor, Spool | Clogging, Backlash, Friction, Clearance |
| [68] | Cylinder | Leakage |
| [69] | Structure | Wear |
| [70] | Cylinder | Leakage, Generic |
| [71] | Cylinder | Leakage |
| [72] | Filter, Motor, Feedback spring, Spool | Clogging, Motor degradation, Backlash, Friction, Clearance |
| [73] | Cylinder, Amplifier | Leakage, Amplifier Fault |
| [74] | Generic | Generic |
| [75] | Cylinder | Leakage |
| [76] | Cylinder | Leakage |
| [77] | Cylinder, Amplifier, Motor | Leakage, Amplifier Fault, Motor Disconnection |
| [78] | Motor, Feedback spring, Mechanism | Motor degradation, Crack, Backlash |
| [79] | Spool, Amplifier | Leakage, NCB shift, Friction, Wear |
| [80] | Cylinder | Leakage |
| [81] | Structure | Crack |
| [82] | Cylinder, Amplifier, Spool | Leakage, NCB shift |
| [83] | Cylinder | Leakage |
| [84] | Feedback spring, Seals | Crack, Wear |

**Table A3.** *Cont.*

| Ref. No. | Focus Area | Faults (Functions) |
|---|---|---|
| [85] | Filter | Clogging |
| [86] | Feedback spring, Motor, Jet pipe, Mechanism | Motor degradation, Crack, Backlash, Deformation |
| [87] | Cylinder | Leakage |
| [88] | Mechanism, Feedback spring, Motor, Jet pipe, Seals, Sensor | Backlash, Crack, Short circuit, Motor degradation, Clogging, Deformation, Wear |
| [89] | Spool, Amplifier | Leakage, NCB shift, Friction, Wear |
| [90] | Generic | Generic |
| [91] | Structure | Wear |

**Table A4.** Exploited signals for each record.

| Ref. No. | Signals |
|---|---|
| [64] | Differential pressure, Servo-valve current, Command, Spool position |
| [65] | Act. position, Spool position, Differential pressure, Servo-valve current |
| [66] | Servo-valve current |
| [67] | Act. position, Spool position, Servo-valve current, Temperature |
| [68] | Command, Act. position, Spool position, Differential pressure |
| [69] | Pressure gain, Null-leakage flow, Lap |
| [70] | Command, Act. position |
| [71] | Command, Act. position, Spool position, Differential pressure |
| [72] | Act. position, Servo-valve current |
| [73] | Command, Act. position |
| [74] | Flow-Pressure coefficient, Spool gain |
| [75] | Act. position |
| [76] | Command, Act. position, Spool position, Differential pressure, Servo-valve current, Temperature, Supply pressure |
| [77] | Command, Act. position, Aerodynamic loads |
| [78] | Command, Act. position, Spool position, Differential pressure, Servo-valve current, Solenoid valve current, Solenoid valve position |
| [79] | Act. position |
| [80] | Act. position, Solenoid valve position |
| [81] | Load cycles |
| [82] | Command, Act. position |
| [83] | Differential pressure, Input voltage |
| [84] | Act. position, Servo-valve current |
| [85] | Act. position, Spool position, Differential pressure, Filter flow-rate, Filter differential pressure, Temperature |
| [86] | Command, Act. position, Spool position, Differential pressure, Servo-valve current, Solenoid valve current, Solenoid valve position |
| [87] | Spool position, Differential pressure, Act. speed |
| [88] | Act. position, Spool position, Differential pressure, Servo-valve current |
| [89] | Act. position |
| [90] | Zero-Bias-Current |
| [91] | Pressure gain, Leakage |

**Table A5.** Records sources names and document type.

| Ref. No. | Source | Doc. Type |
| --- | --- | --- |
| [64] | Annual Forum Proceedings—American Helicopter Society | Conference paper |
| [65] | IEEE Aerospace Conference | Conference paper |
| [66] | IEEE Aerospace Conference | Conference paper |
| [67] | Annual Conference of the Prognostics and Health Management Society | Conference paper |
| [68] | Annual Conference of the Prognostics and Health Management Society | Conference paper |
| [69] | Engineering Failure Analysis | Article |
| [70] | Applied Mathematical Modelling | Article |
| [71] | AIAA Non-Deterministic Approaches Conference | Conference paper |
| [72] | Annual Conference of the Prognostics and Health Management Society | Conference paper |
| [73] | Scientia Iranica | Article |
| [74] | ASME Dynamic Systems and Control Conference | Conference paper |
| [75] | IEEE Access | Article |
| [76] | International Conference on Through-life Engineering Services | Conference paper |
| [77] | Mechanical Systems and Signal Processing | Article |
| [78] | Annual Conference of the Prognostics and Health Management Society | Conference paper |
| [79] | International Conference on Control, Decision and Information Technologies | Conference paper |
| [80] | Proceedings of the Institution of Mechanical Engineers, Part I: Journal of Systems and Control Engineering | Article |
| [81] | IEEE Transactions on Instrumentation and Measurement | Article |
| [82] | IEEE Systems Journal | Article |
| [83] | Proceedings of the Institution of Mechanical Engineers. Part I: Journal of Systems and Control Engineering | Article |
| [84] | Aerospace | Article |
| [85] | Annual Conference of the Prognostics and Health Management Society | Conference paper |
| [86] | International Journal of Prognostics and Health Management | Article |
| [87] | Actuators | Article |
| [88] | Annual Conference of the Prognostics and Health Management Society | Conference paper |
| [89] | IEEE Transactions on Control Systems Technology | Article |
| [90] | IEEE Sensors Journal | Article |
| [91] | Applied Sciences | Article |

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
