# Peer review of "A Systematic Literature Review on PHM Strategies for (Hydraulic) Primary Flight Control Actuation Systems"

_actuators, doi:10.3390/act14080382_

Round 1
Reviewer 1 Report
Comments and Suggestions for Authors
This paper is a literature review on PHM strategies for hydraulic primary flight control actuator systems. The existing works in the past 20 years are comprehensively, systematically, and categorically summarized, which can help both the academy and practitioners to understand the current status. The following concerns should be further addressed:
- The few-shot problem is very unique and remarkable for the flight control problems because the safety concern is the first priority. The authors should make some recommendations for this problem, maybe adversarial generating network is a possible solution.
- The general available sensors that can be used in the flight control PHM strategy should be reasonably summarized, which can provide a realistic background for the following investigators.
Author Response
This paper is a literature review on PHM strategies for hydraulic primary flight control actuator systems. The existing works in the past 20 years are comprehensively, systematically, and categorically summarized, which can help both the academy and practitioners to understand the current status. The following concerns should be further addressed:
The few-shot problem is very unique and remarkable for the flight control problems because the safety concern is the first priority. The authors should make some recommendations for this problem, maybe adversarial generating network is a possible solution.
Thank you for this suggestion. This is indeed a good point. We followed the reviewer suggestion and we added a completely new paragraph in the subsection “Few-shots Phenomenon and Data Imbalance” better discussing the implication and solution related to this challenge (please see lines 776-794). In particular, we have introduced and briefly discussed possible solutions including Generative Adversarial Networks (GANs), along with transfer learning, hybrid modeling and physics-informed techniques. We have added references accondingly.
The general available sensors that can be used in the flight control PHM strategy should be reasonably summarized, which can provide a realistic background for the following investigators.
Following the reviewer’s valuable suggestion, we have added a dedicated paragraph summarizing the current state-of-the-art regarding sensors typically installed on flight control EHAs (Lines 568-584). This addition aims to provide a realistic and informative background for future investigators working on PHM strategies.
However, discussing sensor installations in flight control systems presents notable challenges. Sensor configurations are often considered sensitive and proprietary information by manufacturers and operators, leading to limited public disclosure. Consequently, comprehensive details about sensor types, placements, and specifications are not widely available in open literature. While the overview may not capture all variations across different platforms or manufacturers, it provides a reasonable representation of the sensor landscape relevant to flight control EHAs. We believe this context will assist readers and researchers in understanding the practical constraints and opportunities for PHM implementation.
Overall we have taken this opportunity to review the overall manuscript to further increase the clarity, resulting in minor grammar and formal modifications as highlighted in the new version of the manuscript.
Reviewer 2 Report
Comments and Suggestions for Authors
Please read carefully all my comments and questions before any re-submission of a revised version of your paper.
Page 2 ... the study focused on engines ... What do you mean with "engines" ? The aero-engines / turbofans ? Can you be clearer ? To move what in engines precisely ? The nozzle ? The compressor vanes (IGVs and RCVVs) ? Bleed air valves ? Page 3 ... primary flight controls ... Do you explain anywhere the differences between primary and secondary (and other) flight controls (+ examples) ? This is essential when speaking about their actuators ! Page 8 Have you looked at another information about all these papers: are the authors from industry OR academia (university and research centers RC) ... this is a bit related to your RQ2 but I think this distinction / difference should be more detailed ... Page 9 Do you explain anywhere why you treat only primary flight control applications ? Page 10 EMA for primary flight control applications are probably not for "tomorrow" but for "after tomorrow" use ... could you make a short comment about this longer perspective in your document ? Page 11 ... safety concerns ... along with thermal issues ... Are these the only problems / issues with EMAs ? Can you be more explicit / more complete ? What about lifetime (and wear degradation - of the screw mechanism e.g.) with / of EMA for example ? Page 24 Can you please extend the text of the 1st paragraph on this page ? What about reliability of EMAs ? Lifetime of EMAs ? Why not TODAY for primary flight control surfaces ? Wear problems ? Cost ? EMI issues ? Page 24 ... future research directions of EHA PHM ... Have you had access to military fighters technical studies related to EHA for flight controls ? EF-2000, F-16, Gripen, ... ?
Some small improvements needed (see attached pdf file please).
Author Response
Page 2 ... the study focused on engines ... What do you mean with "engines" ? The aero-engines / turbofans ? Can you be clearer ? To move what in engines precisely ? The nozzle ? The compressor vanes (IGVs and RCVVs) ? Bleed air valves ?
We appreciate this suggestion as the explaination was indeed a bit vague. Further details have been added wherever engines are cited in the text (lines 37, 43-47, 391) explicitly citing the technogoly of choice. In this context, the specific datasets are cited in the introducion in lines 43-47
Page 3 ... primary flight controls ... Do you explain anywhere the differences between primary and secondary (and other) flight controls (+ examples) ? This is essential when speaking about their actuators !
We appreciate the reviewer's observation, as we believe this modification significantly enhances the overall clarity of the manuscript. Detailed new paragraphs have been added at lines 87-105 and 107-131, which introduce primary and secondary flight controls, explains their differences, and references subsequent sections. The paragraph is introduced in lines 74-77 as well. Additionally, it addresses the rationale for our focus on primary flight controls, in response to the comment on Page 9.
Page 8 Have you looked at another information about all these papers: are the authors from industry OR academia (university and research centers RC) ... this is a bit related to your RQ2 but I think this distinction / difference should be more detailed ...
We thank the reviewer for this valuable suggestion. In response, we have expanded the discussion related to the authors’ affiliations in the subsection addressing the research landscape (Figure 7(b)), explicitly differentiating between academic and industrial institutions, as well as drawing some possible causes and effects (lines 424-428). A similar appraoch has been reported in the conclusion section (line 907), reflecting on this unbalance. In fact, this distinction reinforces our observation regarding the limited publicly available industrial involvement and supports the implications discussed in RQ7 about the maturity and application readiness of PHM strategies in this domain.
Page 9 Do you explain anywhere why you treat only primary flight control applications ?
We thank the reviewer for the observation. This point is indeed directly connected to the suggestion previously raised regarding Page 3, which we have addressed in the revised manuscript. Along with the newly added explanation distinguishing between primary and secondary flight controls (lines 87-105 and 107-131), we have now explicitly stated that this review focuses exclusively on primary flight controls (line 126). As mentioned, the substantially different configurations and design requirements of actuators for primary and secondary flight controls would make a unified treatment potentially confusing and counterproductive, especially when discussing PHM strategies. Such an approach would result in a loss of clarity and hinder a consistent understanding of the topic.
This clarification complements the previously included reference to the inclusion/exclusion criteria adopted in our SLR, which explicitly excludes secondary flight control actuators from the scope of the study. Finally, these considerations have been added in the manuscript as well (lines 107-131).
Page 10 EMA for primary flight control applications are probably not for "tomorrow" but for "after tomorrow" use ... could you make a short comment about this longer perspective in your document ? Page 11 ... safety concerns ... along with thermal issues ... Are these the only problems / issues with EMAs ? Can you be more explicit / more complete ? What about lifetime (and wear degradation - of the screw mechanism e.g.) with / of EMA for example ? Page 24 Can you please extend the text of the 1st paragraph on this page ? What about reliability of EMAs ? Lifetime of EMAs ? Why not TODAY for primary flight control surfaces ? Wear problems ? Cost ? EMI issues ?
We appreciate the reviewer's constructive feedback, which has enabled us to incorporate additional content that better contextualizes the impact of EMAs in primary flight control.
Specifically, starting from line 370 to 386, we have expanded the paragraph to include considerations on the long-term adoption prospects of EMAs. Additionally, we have elaborated on known challenges associated with EMA adoption, such as thermal management, single point of failure, wear, and lifetime issues, providing a comprehensive explanation.
A similar approach has been taken for the section on page 24, where the paragraph has been extended to better address the discussion from lines 838 to 846. Relevant references have also been included to support these discussions.
Page 24 ... future research directions of EHA PHM ... Have you had access to military fighters technical studies related to EHA for flight controls ? EF-2000, F-16, Gripen, ... ?
Thank you for this insightful observation. We fully agree that access to technical studies and documentation from advanced military platforms such as the EF-2000, F-16, or Gripen would greatly enhance the depth and specificity of our analysis, particularly in terms of operational reliability, maintenance practices, and PHM system integration. However, as these documents are typically classified or proprietary, the current study remains limited to publicly available literature. We have clarified this boundary in the manuscript to better frame the scope of our review (line 291).
In addition to these comments we have addressed all the other observation found in the PDF document. See changes in red in the abstract (line 15-16), keywords (line 17), section titles (lines 85, 429, 486, 748), landing gear brakes (line 40), Lithium-Ion Batteries (line 43), Maré (line 129), “-“ (line 368), servo valve (line 896), that (Lines 748, 750). Moreover, Figure 8 has been changed along with its caption (line before 464).
We have clarified the paragraph “Demonstration of an Acceptable Return on Investments”, please see lines 863.
Overall we have taken this opportunity to review the overall manuscript to further increase the clarity, resulting in minor grammar and formal modification as highlighted in the new version of the manuscript.
Reviewer 3 Report
Comments and Suggestions for Authors
The authors state that they conduct an in-depth analysis of state-of-the-art Prognostics and Health Management (PHM) strategies for aerospace applications, specifically for flight control systems powered by Electro-Hydrostatic Actuators (EHAs), and that they identify limitations and suggest future research directions based on a Systematic Literature Review (SLR). Furthermore, they claim that the manuscript "provides a solid base on which future PHM strategies could be built on, highlighting the current state-of-the-art as well as the challenges encountered during the development of PHM systems for flight controls powered with EHAs."
While the topic is timely and relevant, I believe this assertion may overstate the scientific and technological impact of the study. The review is based on the analysis of only 28 papers, which seems rather limited given the scope implied by the authors. As such, the conclusions should be interpreted with caution. Instead of presenting the work as a definitive foundation for future PHM development, it would be more appropriate to frame it as an initial outlook or a preliminary assessment of existing trends and gaps in the literature.
Minor comment: Line 378: Typo “,Model” should be corrected.
Author Response
The authors state that they conduct an in-depth analysis of state-of-the-art Prognostics and Health Management (PHM) strategies for aerospace applications, specifically for flight control systems powered by Electro-Hydrostatic Actuators (EHAs), and that they identify limitations and suggest future research directions based on a Systematic Literature Review (SLR). Furthermore, they claim that the manuscript "provides a solid base on which future PHM strategies could be built on, highlighting the current state-of-the-art as well as the challenges encountered during the development of PHM systems for flight controls powered with EHAs." While the topic is timely and relevant, I believe this assertion may overstate the scientific and technological impact of the study. The review is based on the analysis of only 28 papers, which seems rather limited given the scope implied by the authors. As such, the conclusions should be interpreted with caution. Instead of presenting the work as a definitive foundation for future PHM development, it would be more appropriate to frame it as an initial outlook or a preliminary assessment of existing trends and gaps in the literature.
We thank the reviewer for the valuable observation and fully agree that the number of papers identified is relatively limited. However, we would like to emphasize that this number represents the complete body of literature currently available on PHM strategies specifically for flight control systems powered by Electro-Hydraulic Actuators (EHAs), as determined through the rigorous and systematic literature review (SLR) protocol. Indeed, our methodology was carefully designed to capture all relevant publications within this specialized field, applying strict inclusion and exclusion criteria.
As such, while the sample size may appear limited at first glance, it accurately represents the entirety of publicly available scientific literature specifically focused on PHM for flight control EHAs. Therefore, the limited number of studies reflects a genuine research gap rather than any deficiency in our review process.
On the other hand, we appreciate the reviewer’s insightful suggestion and agree that the a revised framing could better reflect the current state-of-the-art and the scope of our contribution.Therefore, in response to this valuable feedback, we have modified the conclusions of the manuscript to clearly frame the work as an assessment that synthesizes the entirety of existing research to date, providing a foundation for future studies while keeping into consideration the limited size of the literature base. (Please see line 877)
Minor comment: Line 378: Typo “,Model” should be corrected.
Thank you for highlighting this typo. We have corrected the subsection title.
Moreover, we have taken this opportunity to review the overall manuscript to furtherly increase the calrity, resulting in minor grammar and formal modification as highlighted in the new version of the manuscfript. (Please see line 429)
Round 2
Reviewer 3 Report
Comments and Suggestions for Authors
The authors have satisfactorily implemented the requested improvements. As such, the manuscript merits publication.